# DREAM: DOMAIN-FREE REVERSE ENGINEERING ATTRIBUTES OF BLACK-BOX MODEL

## ABSTRACT

Deep learning models are usually black boxes when deployed on machine learning platforms. Prior works have shown that the attributes ($e.g.$, the number of convolutional layers) of a target black-box neural network can be exposed through a sequence of queries. There is a crucial limitation that these works assume the dataset used for training the target model to be known beforehand, and leverage this dataset for model attribute attack. However, it is difficult to access the training dataset of the target black-box model in reality. Therefore, whether the attributes of a target black-box model could be still revealed in this case is doubtful. In this paper, we investigate a new problem of Domain-free Reverse Engineering the Attributes of a black-box target Model, called DREAM, without requiring the availability of target model's training dataset, and put forward a general and principled framework by casting this problem as an out of distribution (OOD) generalization problem. At the heart of our framework, we devise a multi-discriminator generative adversarial network (MDGAN) to learn domain invariant features. Based on these features, we can learn a domain-free model to inversely infer the attributes of a target black-box model with unknown training data. This makes our method one of the kinds that can gracefully apply to an arbitrary domain for model attribute reverse engineering with strong generalization ability. Extensive experimental studies are conducted and the results validate the superiority of our proposed method over the baselines.

## 1 INTRODUCTION

With its commercialization, machine learning as a service (MLaaS) is becoming more and more popular, and providers are paying more attention to the privacy of models and the protection of intellectual property. Generally speaking, the machine learning service deployed on the cloud platform is a black box, where users can only obtain outputs by providing inputs to the model. The attributes of the model such as architecture, training set, training method, are concealed by provider. However, if such a deployment is safe? Once the attributes of the model are revealed, it will be beneficial to many downstream attacking tasks, e.g., adversarial example generation (Moosavi-Dezfooli et al., 2016), model inversion (He et al., 2019), etc.

(Oh et al., 2018) has conducted model reverse engineering to reveal model attributes, as shown in the left of Figure 1. They first collect a large set of white-box models which are trained based on the same datasets as the target black-box model, $e.g.$, the MNIST hand-written dataset (Lecun et al., 1998). Given a sequence of input queries, the outputs of white-box models can be obtained. After that, a meta-classifier is trained to learn a mapping between model outputs and model attributes. For inference, outputs of the target black-box model are fed into the meta-classifier to predict model attributes. The promising results demonstrate the feasibility of model reverse engineering.

However, a crucial limitation in (Oh et al., 2018) is that they assume the dataset used for training the target model to be known in advance, and leverage this dataset for meta-classifier learning. In most application cases, the training data of a target black-box model is unknown. When the domain of training data of the target black-box model is inconsistent with that of the set of constructed white-box models, the meta-classifier is usually unable to generalize well on the target black-box model. To verify this point, we train three black-box models with the same architecture on three different datasets, Photo, Cartoon and Sketch(Li et al., 2017), respectively. We use the method in (Oh et al., 2018) to train a meta-classifier on the white-box models which are trained on the Cartoon

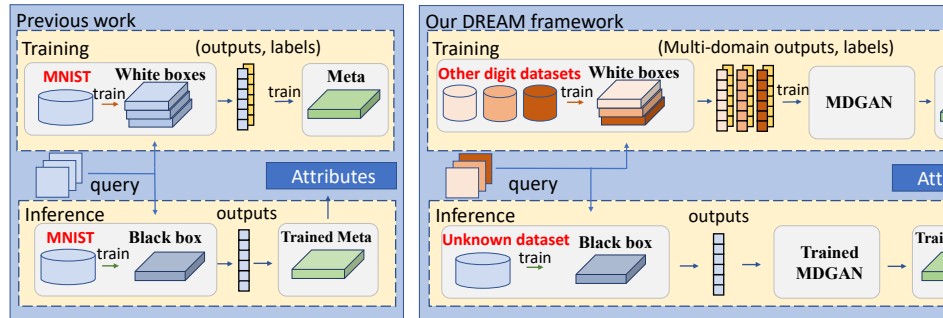

Figure 1: Previous work (left) assumes the dataset used to train the target black-box model is given beforehand, and requires to use the same dataset to train white-box models. Our DREAM framework (right) relaxes the condition that training data of black-box model is no longer required to be available, and proposes a domain-free method to infer attributes of a black-box model. Our idea is casting the problem into an out-of-distribution learning problem, and designing a GAN (Goodfellow et al., 2014) based network (MDGAN) to learn domain invariant features for black-box model attribute inference.

dataset. After that, we use the trained meta-classifier to infer attributes of three black-box models, respectively. As shown in Figure 2, when the training dataset of black-box models and white-box models are the same (i.e., Cartoon), the performance reaches about $80\%$, otherwise, it sharply drops to about $40\%$, close to random guess. The huge gap shows that it is not trivial to investigate model reverse engineering with the assumption of the training dataset of black-box model not available. Furthermore, if the training set for black-box model changes, (Oh et al., 2018) needs to retrain the whole set of white-box models to obtain a promising result, which is extremely time-consuming.

In this paper, we investigate the problem of black-box model attribute reverse engineering, no longer requiring training data of the target model is available, as shown in the right of Figure 1. Obviously, when feeding the same input queries to models with the same architecture but training data of different domains, the output distributions of these models are usually different. Thus, in our problem setting, a key point is how to bridge the gap between the output distributions of white-box and target black-box models, due to the lack of the target model's training data. An ideal meta-classifier should be well trained based on outputs of white-box models, and predict well on outputs of the target black-box model, even if white-box and black-box models are trained using data of different domains.

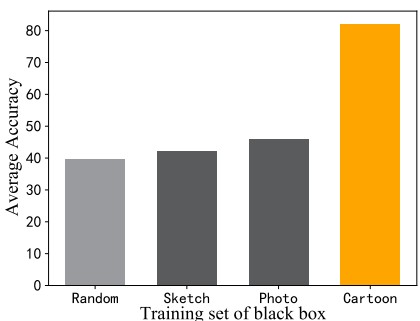

Figure 2: The performance of (Oh et al., 2018) on three datasets.

In light of this, we cast such a problem as an out of distribution (OOD) generalization problem, and propose a novel framework DREAM: Domain-free Reverse Engineering the Attributes of black-box Model. In the field of computer vision, out of distribution generalization learning has been widely studied in recent years (Shen et al., 2021), where its main goal is to learn a model on data of one or multiple domains, and generalize well on data of another domain unseen during training. One kind of mainstream OOD learning approaches is to extract domain invariant features from data of multiple different domains, and utilize the domain invariant features for downstream tasks (Li et al., 2018; Kim et al., 2021; Zhou et al., 2021b). These methods mainly focus on image or video data, and have shown powerful performance. Back to our problem, the black-box models deployed on cloud platform provide their functionality and which categories they can output. Therefore, we can collect data with the same label but different distribution as domains to train white-box models and obtain their probability outputs. since the data we concentrate on is related to the outputs of machine learning models, e.g., probability value, how to design an effective OOD learning method over this type of data has not been explored. To this end, we design a multi-discriminator generative adversarial network (MDGAN) to learn domain invariant features from the outputs of white-box models trained on multi-domain data. Based on learnt domain invariant features, we learn a domain-free reverse

model which can well infer the attributes of a target black-box model trained using data of an arbitrary domain.

Our contributions are summarized as follows: 1) We provide the first study on the problem of domain-free reverse engineering the attributes of black-box models, and cast it as an out of distribution (OOD) generalization problem; 2) We propose a generalized framework, DREAM, which can address the problem of inferring the attributes of a black-box model with an arbitrary training domain; 3) We constitute the first attempt to explore learning domain invariant features from probability representations, in contrast to traditional image representations; 4) We perform extensive experiments and analyze the results, demonstrating the effectiveness of our method.

## 2 RELATED WORK

**Reverse Engineering of Model Attribute.** Its goal is to reveal attribute values of a target model, such as model structure, optimization method, hyperparameters, *etc.* Current research efforts focus on two aspects, hardware (Yan et al., 2020; Hua et al., 2018) and software (Oh et al., 2018; Wang & Gong, 2019). The hardware-based methods utilize information leaks from side-channel (Hua et al., 2018; Yan et al., 2020) or unencrypted PCIe buses (Zhu et al., 2021) to invert the structure of deep neural networks. Software-based methods reveal model attributes by machine learning. (Wang & Gong, 2019) steals the trade-off weight of loss function and the regularization term. They derive over-determined linear equations and solve the hyperparameters by least-square method. KENNEN (Oh et al., 2018) prepares a set of white-box models, and then trains a meta-classifier to build a mapping between model outputs and their attributes. It is the most related work to ours. However, a significant difference is that, Oh et al. (2018) requires the data used to train the target black-box model to be given beforehand, while our method relaxes this condition, *i.e.*, we no longer require the training data of target model to be available. Thus, we attempt to solve a more practical problem.

**Model Functionality Extraction.** It aims to train a clone model that has similar model functionality to that of the target model. To achieve this goal, many works have been proposed in recent years (Orekondy et al., 2019; Truong et al., 2021; Papernot et al., 2017). (Orekondy et al., 2019) uses an alternative dataset collected from Internet to query the target model. (Papernot et al., 2017) assumes part of dataset is known, and then presents a dataset augmentation method to construct the dataset for querying the target model. Moreover, data-free extraction methods (Kariyappa et al., 2021; Truong et al., 2021) query a target model through data generated by a generator, and use zero-order gradient approximation to approximate the gradient of the target model. Different from the methods mentioned above, our goal is to infer the attributes of a black-box model, rather than stealing the model function.

**Membership Inference.** Its goal is to determine whether a sample belongs to the training set of a model (He et al., 2020; Choquette-Choo et al., 2021; Rezaei & Liu, 2021). Although inferring model attribute is different from the task of membership inference, the technique in Oh et al. (2018) is actually similar to those of membership inference attack. However, as stated aforementioned, when the domain of training data of the target black-box model is inconsistent with that of the set of white-box models, the method is usually unable to generalize well because of the OOD problem.

**OOD Generalization.** The goal of OOD Generalization is to deal with the inevitable shifts from a training distribution to an unknown testing distribution (Shen et al., 2021). Existing methods mainly fall into three categories: domain generalization(Kim et al., 2021; Li et al., 2018; Zhou et al., 2021b;a; Hu et al., 2020), causal learning (Arjovsky et al., 2019; Creager et al., 2021; Krueger et al., 2021; Mahajan et al., 2021) and stable learning (Shen et al., 2020; Kuang et al., 2020; Zhang et al., 2021; Kuang et al., 2018). Domain generalization attempts to learn invariant representations among different domains. The work closest to us is ADA (Ganin et al., 2016) which uses an adversarial strategy between a feature extractor and a discriminator to learn domain invariant features. ADA is designed for domain adaptation task (only two domain). However, we aim to solve a domain generalization problem that handle more than two domains, single discriminator cannot learn domain invariant feature between multiple domains. Causal learning and stable learning aim to search for causal features to ground-truth labels from data and filter out label-unrelated features. The former makes existing causal features invariant, while the later focuses on the effective features strongly related to labels by reweighting attentions. The above methods mainly focus on image or video. How to design an effective OOD learning method for attribute inference of black-box model has not been explored so far.

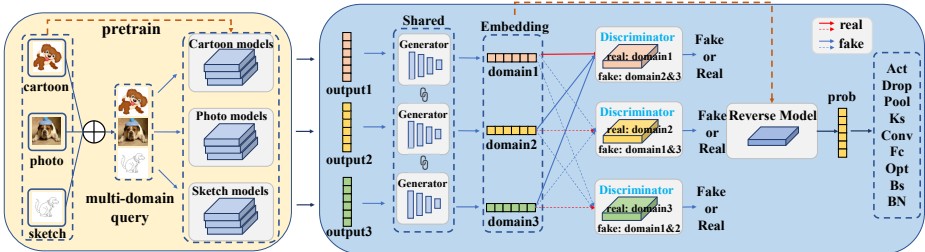

Figure 3: An illustration of our DREAM framework. 1) In the left part, we prepare a large number of white-box models from different domains. In each domain, models have various combinations of attributes, then we input multi-domain queries into each white-box model to obtain a multi-domain model output. 2) In the right part, the core idea is designing a multiple-discriminator GAN network (MDGAN) to learn domain invariant features from the outputs of white-box models trained on multi-domain data. After that, a domain-free reverse model is learnt based on domain invariant embeddings, and is used to infer the attributes of black-box model with arbitrary domains.

## 3 PROPOSED METHOD

### 3.1 PRELIMINARIES

**KENNEN (Oh et al., 2018).** Given a black-box model $B$, model attribute reverse engineering in (Oh et al., 2018) aims to build a meta-classifier $\Phi : B \rightarrow A$, where $A$ is the set of model attributes including model architecture, optimizer and training hyperparameters, *etc.* Concretely, they firstly collect a large set of white-box models $\mathcal{F}$ containing different attributes combination, and train these white-box model based on the same training data $\mathcal{D}$ as that of the target black-box model. Then outputs $O$ is obtained by querying these white-box models with a sequence of input images. Finally they build a mapping $\Phi$ from outputs $O$ to model attributes $A$, realized as a meta-classifier $\Phi$. At the inference phase, the meta-classifier takes outputs from target model as input and predicts the corresponding attributes.

**Problem Formulation.** As aforementioned, there is a strict constraint in (Oh et al., 2018) that they assume the training dataset $\mathcal{D}$ of the target model to be given in advance, and leverage $\mathcal{D}$ for learning meta-classifier $\Phi$. In most scenarios, especially on public machine learning platforms, it is difficult to access the training data of a target black-box model, which significantly limits the applications of (Oh et al., 2018). To mitigate this problem, we provide a new problem setting by relaxing the above constraint, *i.e.*, we no longer require the training data $\mathcal{D}$ of the target black-box model to be available. Thus, our goal is to learn a domain-free reverse classifier $\Phi$ that is trained based on outputs of white-box models $\mathcal{F}$, and predict well for the target black-box model, even if white-box and black-box models are built based on training data of different domains.

### 3.2 DREAM FRAMEWORK

To perform domain-free black-box model attribute reverse engineering, we cast this problem into an out-of-distribution (OOD) generalization learning problem, and propose a novel framework DREAM, as shown in Figure 3. Our DREAM framework consists of two parts: In the left part of Figure 3, we train a number of white-box models with training sets from different domains. Models of each domain are enumerated with various model attributes. All of these models constitute a model set covering different domains (please refer to Sect. 4.1 for more details). Next, we prepare queries as input to these models. For each domain, we sample an equal number of images from the corresponding dataset, and concatenate them as a batch of queries. These queries are sent to each model, and outputs of the model are fed into the other module of our DREAM framework, as shown in the right part of Figure 3. The core idea is to design a multi-discriminator generative adversarial network (MDGAN) to learn domain invariant features, where MDGAN consists of multiple discriminators corresponding to different domains and one generator across multiple domains. The generator aims to learn domain invariant features, and each discriminator intends to make the learnt feature distributions of other domains to fit that of the domain itself. In this way, the generator is capable of learning domain invariant features. Based on the learnt domain invariant features, we can learn a domain-free reverse model to infer the attributes of a black-box model with an arbitrary domain.

### 3.3 MULTI-DOMAIN OUTPUT PREPARATION

The multi-domain output can be taken as a representation of a white-box model, and is fed into MDGAN to learn domain invariant features. Specifically, we sample an equal number of images from the dataset of each domain to obtain a query set $Q = \{q_j\}_{j=1}^{N}$, where $N$ is the number of queries. We denote training model set from each domain as $\mathcal{F} = [\boldsymbol{f^1}, \boldsymbol{f^2}, ..., \boldsymbol{f^m}]$, where $\boldsymbol{f^i}$ consists of $K$ models of $i^{th}$ domain. Then, we input each query $q_j \in Q$ into models $\boldsymbol{f^i}$ of $i^{th}$ domain to get an output $O_j^i \in R^{K \times C}$, where $O_j^i$ represents $K$ outputs of $i^{th}$ domain for a query. We obtain $O^i \in R^{K \times CN}$ by concatenating $N$ outputs. Finally, we derive multi-domain outputs as $O = [O^1, ..., O^m] \in \mathbb{R}^{m \times K \times CN}$.

The core idea of MDGAN is to learn embeddings for each domain by a parameter sharing generator, and make the distributions of different domains as close as possible by multiple discriminators.

### 3.4 MULTI-DISCRIMINATOR GAN (MDGAN)

After preparing multi-domain outputs, we devised a GAN based network, MDGAN, to learn domain invariant features from the outputs of white-box models trained on multi-domain representation. To better present, we take Figure 4 to illustrate the idea behind MDGAN. Assume there are two kinds of inputs, $O^1$ and $O^2$, from two domains. When feed them into the generator $G$, we can obtain the corresponding embeddings $z^1$ and $z^2$, respectively. After that, we feed $z^1$ and $z^2$ to the discriminator $D_1$, where $D_1$ is expected to output a "real" label for $z^1$ and output a "fake" label for $z^2$. By jointly training $G$ and $D_1$ based

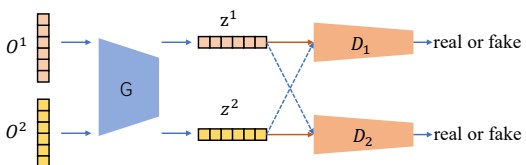

Figure 4: An example to illustrate the idea behind MDGAN.

on a minmax optimization, the distribution of $z^2$ is expected to move towards that of $z^1$. In the meantime, we also feed $z^1$ and $z^2$ to the discriminator $D_2$. Differently, $D_2$ is expected to output a "real" label for $z^2$ and output a "fake" label for $z^1$. By jointly training $G$ and $D_2$, the distribution of $z^1$ is expected to move towards that of $z^2$. In this way, $z^1$ and $z^2$ generated by the generator $G$ become domain invariant representations.

Formally, we define $G(O; \theta_g) : O \rightarrow z$. The generator $G$ sharing with parameter $\theta_g$ across domains maps multi-domain outputs $O$ into the latent feature $z$. We also define $m$ discriminators $\{D^i(z; \theta_d^i)\}_{i=1}^{m}$. Each discriminator $D^i(z) : z \rightarrow [0, 1]$ outputs a scalar representing the probability that $z$ comes from the $i^{th}$ domain rather than others. For $D^i(z)$, we treat the correct label of an embedding in the $i^{th}$ domain, *i.e.*, $O^i$, as *True*, while others as *False*. Then we divide multi-domain outputs into two groups, $\{O_T^i\}$ and $\{O_F^i\}$, which are defined as:

$$\{O_T^i\} = \{O^i\}; \; \{O_F^j\} = \{O^j | j \neq i\}; \; \bigcup_{j \neq i} \{O_F^j\} \cup \{O_T^i\} = O. \tag{1}$$

The training goal of $D^i$ is to maximize the probability of assigning the correct label to features both from the $i^{th}$ domain and other domains, while the generator $G$ is trained against the discriminator to minimize $log(1 - D(G(x)))$. In other words, it is a min-max game between the $i^{th}$ discriminator $D^i$ and generator $G$ with a value function $V$, formulated as:

$$\min_{G} \max_{D^i} V(D^i, G) = \mathbb{E}_{x \sim \{O_T^i\}}[log D^i(G(x))] + \sum_{j \neq i} \mathbb{E}_{x \sim \{O_F^j\}}[log(1 - D^i(G(x)))]. \tag{2}$$

During optimizing the min-max adversarial loss for $G$ and $D^i$, the distributions of model outputs from the $i^{th}$ domain and other domains become closer. After $G$ and all $D$ are well trained, $G$ will embed multi-domain model outputs into an invariant feature space, where each discriminator cannot figure out which domain the outputs of white-box models are from.

### 3.5 DOMAIN-FREE REVERSE MODEL

Then, we use the domain-free reverse classifier to classify the domain invariant features produced by the generator. We denote features $z$ produced by $G(O; \theta_g)$ as

$$z = [G(O^1); G(O^2); ...; G(O^m)] \in \mathbb{R}^{m \times K \times d'}. \tag{3}$$

Where $d'$ is the number of feature dimensions. We define the domain-free reverse classifier as $\Phi(z; \theta_c)$ parameterized by $\theta_c$. We obtain probability $p(z^i)$ for each possible model attribute as:

$$p(z^i) = softmax(\Phi(z^i)) = \frac{exp\{\Phi(z^i)\}}{\sum_{i=1}^m exp\{\Phi(z^j)\}}. \tag{4}$$

The target is to minimize the cross entropy between the predicted $p(z^i)$ and ground-truth of model attribute values $y$:

$$\min_\Phi \mathbb{E}_{z \sim G(O)} \left[ \sum_{i=1}^C -y^i log(p(z^i)) \right] = \min_\Phi \mathbb{E}_{z \sim G(O)} \left[ -y^T log(p(z)) \right]. \tag{5}$$

At inference phase, given the same queries as the white-box model, the outputs of a black-box model from an unknown domain are fed into the generator $G$, and then the output of $G$ is fed into the reverse classifier $\Phi$, achieving domain-free prediction of black-box model attributes.

## 3.6 Overall Model and Training Strategy

After introducing all the components, we give the final loss function based on Eq. 2 and 5 as:

$$\min_{G,\Phi} \max_{D^i, 1 \le i \le m} V(D^i, G) = \mathbb{E}_{x \sim \{O_T^i\}} \left[ log D^i(G(x)) \right] + \sum_{j \ne i} \mathbb{E}_{x \sim \{O_F^j\}} \left[ log(1 - D^i(G(x))) \right]$$
$$+ \lambda \mathbb{E}_{z \sim G(O)} \left[ -y^T log(p(z)) \right].$$

where $\lambda$ is a trade-off parameter. We observe if we firstly train MDGAN, and then optimize domain-free reverse classifier, the generator of MDGAN will converge to a trivial solution. In other words, it tends to produce identical features, resulting in a reverse model which cannot predict model attributes correctly. Thus, we design a training strategy: we first optimize all discriminators $D^i$, and then jointly optimize the generator and the domain-free reverse classifier. We repeat the above processes, until the algorithm converges. The proposed training strategy is represented in Algorithm 1 of Appendix A.3.

## 4 Experiments

### 4.1 Dataset Construction

Following (Oh et al., 2018), we train a number of models which are constructed by enumerating all possible attribute values. The details of the attributes and their values are shown in Table 1. The number of models with all possible combinations of the attributes is $5,184$. We also initialize each model with random seeds from $0$ and $999$, yielding 5,184,000 unique white-box models. For each domain, we randomly sample and train 10,000 white-box models from 5,184,000 models. Then we ~~also~~ sample 5000, 1000, 1000 from 10,000 white-box models as the training set, validation set, and testing set. Next, we introduce the details of our datasets.

**PACS-modelset**. PACS is an image dataset that has been widely used for OOD learning (Li et al., 2017). In this experiment, we use it for evaluating our domain-free black-box model attribute inference framework DREAM. We utilize three domains, including Photo (1,670 images), Cartoon (2,344 images) and Sketch (3,929 images), to construct our dataset. In our dataset, each domain contains 7 categories. For each domain we train 10,000 models and we combine them as PACS-modelset (30,000 models in total).

**MEDU-modelset**. MEDU is a set of hand-written digit recognition dataset, with 4 domains collected from MNIST (Lecun et al., 1998), USPS (Hull, 1994), DIDA (Kusetogullari et al.) and

Table 1: Attributes and the corresponding values.

| Attribute | Values |
|---|---|
| #Activation | ReLU, PReLU, ELU, Tanh |
| #Dropout | Yes, No |
| #Max pooling | Yes, No |
| #Batchnorm | Yes, No |
| #Kernel size | 3, 5 |
| #Conv layers | 2, 3, 4 |
| #FC layers | 2, 3, 4 |
| #Optimizer | SGD, ADAM, RMSprop |
| #Batch size | 32, 64, 128 |

EMNIST (Cohen et al., 2017). Each domain contains different styles of hand-written digit from 0 to 9. We train 40,000 models as MEDU-modelset and each domain contains 10,000 models.

Table 2: Model attribute classification accuracy (%) on PACS-modelset. **Red** and blue indicate the best and second best performance, respectively.

| | Method | Attributes | | | | | | | | | Avg |
|---|---|---|---|---|---|---|---|---|---|---|---|
| | | #act | #drop | #pool | #ks | #conv | #fc | #opt | #bs | #bn | |
| | Random | 25.00 | 50.00 | 50.00 | 50.00 | 33.33 | 33.33 | 33.33 | 33.33 | 50.00 | 39.81 |
| P | SVM | 37.80 | 50.30 | 54.80 | 53.60 | 34.00 | 36.60 | 37.00 | 45.70 | 58.80 | 45.40 |
| | KENNEN* | 39.07 | 50.68 | 59.42 | 61.31 | 36.18 | 39.33 | 37.88 | 44.16 | 59.74 | 47.53 |
| | SelfReg | 25.58 | 52.26 | 54.98 | 50.18 | 34.12 | 35.25 | 34.61 | 33.78 | 50.76 | 41.28 |
| | MixStyle | 39.63 | 53.23 | 61.83 | 59.44 | 35.66 | 38.75 | 37.89 | 43.75 | 57.09 | 47.47 |
| | MMD | 38.88 | 54.70 | 60.46 | 56.54 | 35.38 | 36.66 | 35.66 | 40.50 | 61.04 | 46.65 |
| | **DREAM** | **43.84** | **59.19** | **66.09** | **64.24** | **39.59** | **42.04** | **40.49** | **47.83** | **68.12** | **52.38** |
| C | SVM | 25.80 | 49.20 | 50.70 | 55.80 | 37.20 | 38.10 | 30.80 | 42.30 | 65.30 | 43.91 |
| | KENNEN* | 32.99 | 52.50 | 54.23 | 56.57 | 37.19 | **40.53** | 33.47 | 37.17 | **68.39** | 45.89 |
| | SelfReg | 25.97 | 51.42 | 56.20 | 50.03 | 35.04 | 35.52 | 36.09 | 35.58 | 56.17 | 42.44 |
| | MixStyle | 32.10 | 50.76 | 55.44 | 54.18 | 36.18 | 37.87 | 34.65 | 38.69 | 60.26 | 44.46 |
| | MMD | 29.56 | 53.02 | 54.70 | 53.82 | 35.38 | 36.36 | 35.98 | 37.24 | 57.58 | 43.75 |
| | **DREAM** | **37.53** | **55.89** | **61.18** | **57.32** | **38.58** | 39.60 | **38.32** | **45.01** | 65.16 | **48.73** |
| S | SVM | 23.80 | 47.60 | 47.40 | 45.80 | 33.80 | 34.50 | 31.80 | 34.30 | 53.10 | 39.12 |
| | KENNEN* | 34.64 | 50.10 | 53.07 | 52.01 | 34.61 | 37.11 | 35.78 | 37.04 | 55.27 | 43.29 |
| | SelfReg | 27.07 | 54.32 | 51.39 | 53.07 | 36.99 | 36.82 | 35.47 | 34.17 | 61.80 | 43.46 |
| | MixStyle | 37.78 | 51.71 | 54.16 | 53.60 | 34.53 | 36.16 | 36.36 | 36.02 | 59.42 | 44.42 |
| | MMD | 31.96 | 52.94 | 56.84 | 52.78 | 38.18 | 38.20 | 36.20 | 35.92 | 57.56 | 44.51 |
| | **DREAM** | **42.24** | **55.68** | **61.82** | **58.34** | **39.55** | **38.39** | **38.51** | **41.39** | **74.39** | **50.03** |

In the experiment, we set the number of queries $N$ to 100. We use Adam (Kingma & Ba, 2014) as the optimizer, where the learning rate $\alpha$ is set to $10^{-5}$ for the generator and discriminators, and the learning rate $\beta$ is set to $10^{-4}$ for the reverse model. The batch size $b$ is set as 100. The trade-off parameter $\lambda$ is tuned from $\{0.001, 0.01, 0.1, 1, 10\}$ based on the validation set. Parameter sensitive analysis can be found in Appendix. In addition, the generator and discriminators are implemented as a two-layer MLP, respectively, where ReLU is used as the non-linear activation function. All experiments are conducted on 4 NVIDIA RTX 3090 GPUs, PyTorch 1.11.0 platform.

We compare our DREAM with 6 baselines including Random choice, SVM, KENNEN (Oh et al., 2018), SelfReg (Kim et al., 2021), MixStyle (Zhou et al., 2021b), MMD (Li et al., 2018). To compare fairly, we select a variant of KENNEN (denoted as KENNEN*) taking fixed queries as input, which is the same as ours. Moreover, we also take three typical OOD generalization methods, SelfReg, MixStyle and MMD, as baselines to verify the effectiveness of our proposed MDGAN network for learning domain invariant features. SelfReg aims to draw samples of similar categories between all domains closer and samples of different categories farther; MixStyle captures style information of images by the CNN layer, and it performs style mixing at the layer; MMD adopts maximum mean discrepancy loss between two domains. To apply OOD baselines, we first take probabilities as input to learn invariant features by them, and then adopt a MLP on these features to predict model attributes. In addition, we take SVM as a basic baseline without considering different domain outputs.

We adopt the "leave-one-domain-out" scheme to split the source and target domains. For each dataset, we in turn take one domain as the target domain and the rest domains as source domains. We run the experiment 10 trials and report the average accuracy on each split.

## 4.2 EXPERIMENTAL RESULTS AND ANALYSIS

**Overall Performance.** Table 2 and 3 report the overall performance of different methods on the PACS-modelset and MEDU-modelset, respectively. The left-most column in each table indicates the target domain (the rest ones are source domains). The performance achieved by our proposed DREAM is better than that of all baselines in terms of the average result of models attributes. For individual attribute, our method outperforms other methods in most of the cases. Our method is better than KENNEN, which illustrates our method benefits from learning domain invariant features and learning domain-free reverse model. Moreover, our method achieves better performance than the three OOD learning methods, which indicates it is necessary to design new methods to extract domain invariant features for model attribute inference of black-box models.

Table 3: Model attribute classification accuracy (%) on MEDU-modelset. **Red** and blue indicate the best and second best performance, respectively.

| | Method | Attributes | | | | | | | | | Avg |
|---|---|---|---|---|---|---|---|---|---|---|---|
| | | #act | #drop | #pool | #ks | #conv | #fc | #opt | #bs | #bn | |
| | Random | 25.00 | 50.00 | 50.00 | 50.00 | 33.33 | 33.33 | 33.33 | 33.33 | 50.00 | 39.81 |
| M | SVM | 45.60 | 49.40 | 62.90 | 59.20 | 38.80 | 40.10 | 35.50 | 35.00 | 75.30 | 49.09 |
| | KENNEN* | 51.18 | 50.67 | 62.99 | 57.36 | 38.32 | 35.84 | 41.57 | 35.75 | 77.87 | 50.17 |
| | SelfReg | 28.00 | 53.57 | 53.43 | 50.78 | 35.97 | 36.39 | 35.98 | 36.23 | 53.96 | 42.70 |
| | MixStyle | 50.27 | 51.72 | 62.66 | 57.32 | 37.88 | 36.34 | 43.11 | 38.00 | 82.61 | 51.10 |
| | MMD | 44.57 | 59.67 | 66.37 | 57.27 | 39.63 | 37.27 | 42.10 | 37.60 | 81.37 | 51.76 |
| | **DREAM** | 51.01 | 62.32 | 64.28 | 58.39 | 40.96 | 38.11 | 45.37 | 38.96 | 81.99 | 53.49 |
| E | SVM | 40.00 | 48.70 | 69.20 | 51.60 | 40.20 | 36.90 | 35.80 | 30.10 | 79.90 | 48.04 |
| | KENNEN* | 45.66 | 51.01 | 65.26 | 53.25 | 40.28 | 36.35 | 41.96 | 36.16 | 81.30 | 50.14 |
| | SelfReg | 27.29 | 52.83 | 53.32 | 52.85 | 33.68 | 35.05 | 35.26 | 35.32 | 53.74 | 42.15 |
| | MixStyle | 43.68 | 51.35 | 67.87 | 57.15 | 42.50 | 39.30 | 42.10 | 38.79 | 82.46 | 51.69 |
| | MMD | 42.03 | 58.43 | 66.27 | 60.80 | 40.80 | 38.67 | 40.00 | 39.97 | 84.00 | 52.33 |
| | **DREAM** | 45.55 | 64.98 | 74.16 | 60.71 | 44.45 | 42.45 | 47.37 | 41.03 | 91.00 | 56.86 |
| D | SVM | 45.00 | 47.80 | 54.60 | 45.50 | 29.40 | 37.60 | 43.30 | 36.50 | 63.70 | 44.82 |
| | KENNEN* | 42.73 | 52.06 | 55.27 | 52.02 | 34.89 | 38.90 | 38.98 | 36.27 | 54.97 | 45.12 |
| | SelfReg | 26.31 | 54.29 | 53.23 | 52.33 | 34.96 | 35.72 | 36.49 | 35.39 | 59.11 | 43.09 |
| | MixStyle | 45.26 | 52.32 | 55.91 | 51.39 | 34.22 | 38.70 | 38.31 | 38.03 | 57.44 | 45.73 |
| | MMD | 39.00 | 59.20 | 59.63 | 55.93 | 35.93 | 38.33 | 37.93 | 37.50 | 54.40 | 46.43 |
| | **DREAM** | 49.63 | 64.50 | 59.30 | 57.13 | 39.52 | 44.59 | 42.09 | 40.19 | 59.68 | 50.74 |
| U | SVM | 43.40 | 50.50 | 47.60 | 52.50 | 30.30 | 32.30 | 41.00 | 36.60 | 49.40 | 42.62 |
| | KENNEN* | 43.38 | 50.88 | 51.41 | 53.19 | 36.35 | 35.59 | 36.66 | 34.56 | 55.62 | 44.18 |
| | SelfReg | 26.81 | 52.16 | 55.46 | 52.47 | 36.18 | 36.43 | 36.53 | 35.90 | 55.34 | 43.03 |
| | MixStyle | 41.05 | 53.80 | 50.49 | 52.93 | 35.26 | 33.68 | 36.92 | 34.75 | 59.34 | 44.25 |
| | MMD | 39.33 | 55.87 | 52.67 | 53.23 | 39.20 | 34.33 | 35.90 | 36.90 | 60.73 | 45.35 |
| | **DREAM** | 42.34 | 58.72 | 58.58 | 54.41 | 37.90 | 37.81 | 40.42 | 38.36 | 63.39 | 47.99 |

What is more, we observe that DREAM cannot outperform other baselines in some cases. The reasons might be: 1) DREAM vs. OOD learning baselines. As we have mentioned, the OOD learning methods aim to learn a domain invariant space from different domains. Once features of different domains are excessively pulled close, the classification accuracy would be influenced. Thus, the trade-off between invariant space learning and classification learning is vital for the performance of reverse engineering. Moreover, the trade-off for each attribute is not identical. Taking MMD in Table 3 E as an example, the best trade-off hyperparameter conduces to predict attribute #ks better than other attributes. Similarly, the best trade-off hyperparameter of DREAM conduces to better predict attributes except for #ks. 2) DREAM vs. KENNEN and SVM. Our proposed DREAM has stronger ability to fit complicated data, while SVM and KENNEN (only a shallow MLP) are weaker in the scenery of complicated data. In the scenery of easier cases, e.g., #act, #ks, #fc in M of MEDU (shown in Table 3), DREAM is more likely to overfit due to more parameters, degrading the performance of our method. However, our method generally performs better than SVM and KENNEN in most cases.

**Visualization of Generated Feature Space**. To further verify the effectiveness of our proposed method, we utilize t-SNE (Van der Maaten & Hinton, 2008) to visualize samples in the domain invariant feature space learnt by the generator G in MDGAN. The visualization is carried out on PACS-modelset. We take C (cartoon) and P (photo) as source domains to train white-box models, and use S (sketch) as the unseen target domain to train black-box model. As shown in Figure 5 a), samples from the three different domains are grouped into individual clusters at the 1st epoch. This illustrates their distributions are indeed different in the beginning. Distributions of source domains (C and P) become closer from epoch 1 to 5. Then, our method embeds features from the unseen domain (S) and the samples from the target domain also become closer to the source domains at the 10th epoch, indicating that our generator is able to generalize an unseen domain into the feature space where the source domains are in. Finally, both source and target domains are transformed into an invariant feature space. For MMD and Mixstyle in Figure 5 b) and Figure 5 c), the distributions of features does not become closer as the training proceeds. For SelfReg in Figure 5 d), the features are pulled closer to some extent from epoch #1 to epoch #10, and part of samples from the unseen target

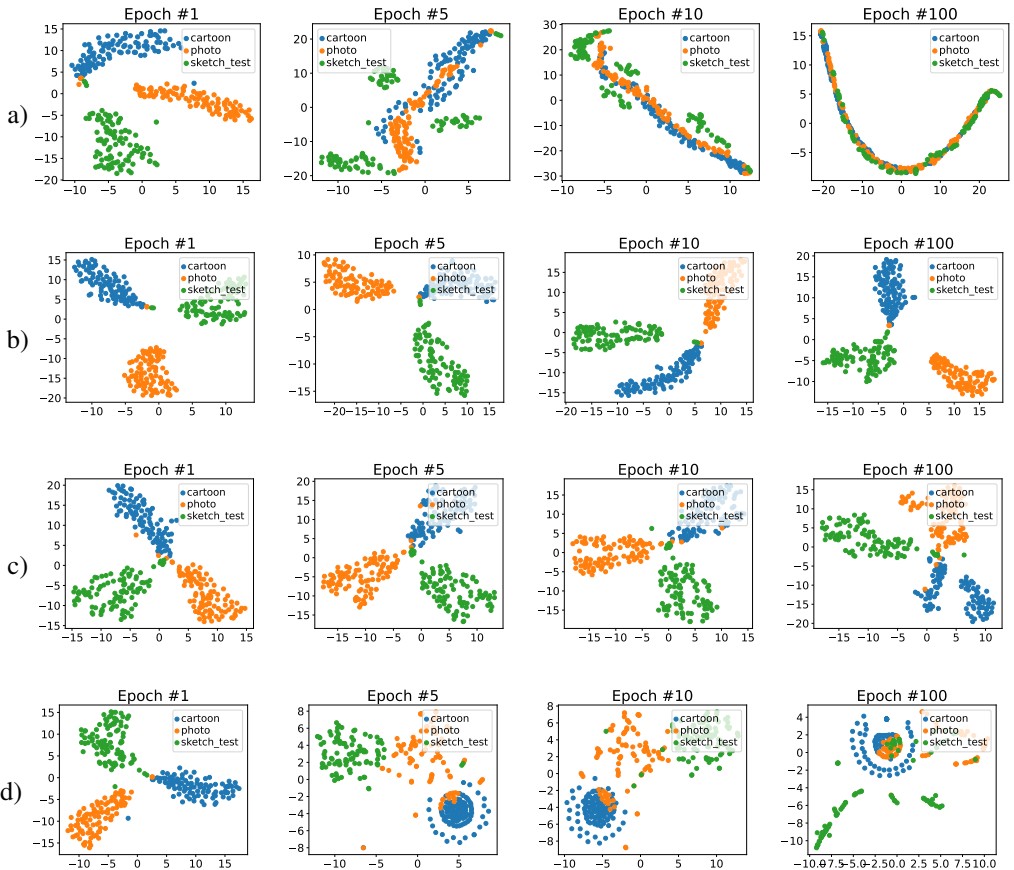

Figure 5: T-SNE visualization of features of different domains produced by a) DREAM, b) MMD, c) MisStyle and d) SelfReg on PACS-modelset.

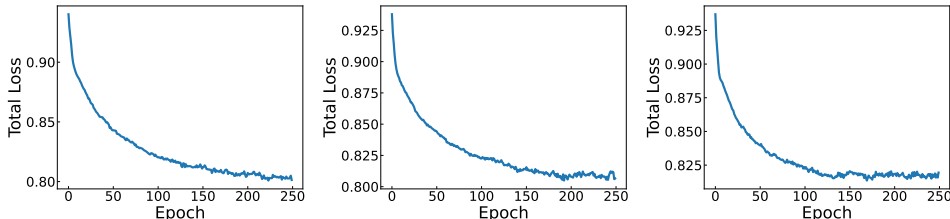

Figure 6: Convergence analysis. Loss curves of meta-classifier (Eq. 5) in the training phase on the three splits P, C, and S (from left to right) demonstrate the convergence of our method.

domain S indeed become closer to the source domains at the $100^{th}$ epoch. However, these feature distributions are not sufficiently tight.

**Convergence Analysis** We study the convergence of our algorithm on the PACS-modelset. The curves of the meta-classifier's loss in the training phase are shown in Figure 6. For all the three splits of domains (left to right), the loss decreases as the training proceeds and finally levels off.

# 5 CONCLUSION

In this paper, we studied the problem of domain-free reverse engineering towards the attributes of black-box model with unknown domain data, and cast it as an OOD generalization problem. We proposed a new framework, DREAM, which can predict the attributes of a black-box model with an arbitrary training domain, and devised a new GAN based network to learn domain invariant features in the scenario of attribute inference of black-box mode. Extensive experimental results demonstrated the effectiveness of our method.

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

## A APPENDIX

### A.1 DETAILS OF CONSTRUCTED MODELSET

We construct two modelsets (PACS-modelset and MEDU-modelset) by enumerating combinations of attribute values. The architecture of each model in modelsets follows the scheme: $N$ convolution layers, $M$ fully-connected layers and a linear classifier. Each convolution layer contains a $k \times k$ convolution, an optional batch normalization, an optional max-pooling and a non-linear activation function in sequence, where $k$ is the kernel size. Each fully-connected layer consists of a linear transformation, a non-linear activation and an optional dropout in sequence. We set the dropout ratio to $0.1$ in our experiments. When training models, optimizers are selected from {SGD, ADAM, RMSprop} with a batch size 32, 64 or 128, respectively.

### A.2 DETAILED IMPLEMENTATION OF MDGAN AND META-CLASSIFIER

The MDGAN is composed of a generator and multiple discriminators. The generator consists of two linear layers with ReLU activation. The dimension of the input layer of the generator is determined by the query number $N$ and class category number $C$. In the experiment of recognizing handwritten digits, the input dimension is 1000 ($N = 100$, $C = 10$). In the case of PACS dataset, the input dimension is 700 ($N = 100$, $C = 7$), and the output dimension of the successive two layers is respectively 500 and 128. Each discriminator consists of three linear layer, with ReLU activation and a final Sigmoid activation. The output dimension of layers are 512, 256, 1 respectively. There are 9 meta-classifiers as total. Each meta-classifier is composed of two layer MLP with dimension of 128, 64, and the length of attribute values.

### A.3 TRAINING STRATEGY ALGORITHM

The training strategy of DREAM is shown in Algorithm 1.

### A.4 EXPERIMENTS ON DIFFERENT TRAINING AND TESTING ATTRIBUTES

We study the case that the white-box model and the black-box model to be inferred have completely different attributes. As we mentioned in Section 4.1, there are $5,184$ combinations of model attributes in total. We randomly sample 3000, 1000, 1000 as training, validation and testing sets. None of the models has identical attributes. As shown in Table 4, DREAM consistently outperforms other baselines on the above setting.

---

**Algorithm 1:** Training Strategy

---

**Input:** Batch size $b$, learning rate $\alpha$, $\beta$, multi-domain model outputs $O$, trade-off scalar $\lambda$

**Output:** Generator $G$, meta-classifier $\Phi$, discriminators $\{D^i\}_{i=1}^{m}$

**Initialize:** Initialize parameter $\theta_g$ of generator $G$, parameter $\theta_d^i$ of discriminators $\{D^i\}_{i=1}^{m}$ and parameter $\theta_c$ of domain-free meta-classifier $\Phi$ with normal distribution

**while** *classifier $\Phi$ not converges* **do**

    Random sample $b$ samples $O_b^i$ from outputs $O^i$ in each domain

    **for** $i = 1, ..., m$ **do**

        Take samples in the $i^{th}$ domain as *True* samples $X = O_b^i = \{x^1, x^2, ..., x^b\}$

        **for** $j = 1, ..., m \ and \ j \neq i$ **do**

            Take samples in the $j^{th}$ domain as *False* samples $\bar{X}_j = O_b^j = \{\bar{x}_j^1, \bar{x}_j^2, ..., \bar{x}_j^b\}$

        **end**

        Update the discriminator $D^i$ by gradient descent:

        $\theta_d^i := \theta_d^i - \alpha \nabla_{\theta_d^i} \left\{ \sum_{k=1}^{b} \left[ logD(G(x^k)] + \sum_{j \neq i} \left[ \sum_{k=1}^{b} log(1 - D(G(\bar{x}_j^k))) \right] \right] \right\}$

    **end**

    Construct $X_{all} = X \cup \bar{X} = \{x_{all}^1, x_{all}^2, ..., x_{all}^{bm}\}$ and $Z_{all} = G(X_{all}) = \{z_{all}^1, z_{all}^2, ..., z_{all}^{bm}\}$

    Set the corresponding labels as $Y_{all} = \{y_{all}^1, y_{all}^2, ..., y_{all}^{bm}\}$

    Calculate gradient of $\theta_c$ and $\theta_g$ by:

    $\boldsymbol{grad_c} = \nabla_{\theta_c} \sum_{k=1}^{bm} \left[ -{y_{all}^k}^T log(p(z_{all}^k)) \right]$

    $\boldsymbol{grad_g} = \nabla_{\theta_g, \theta_c} \left\{ \sum_{j \neq i} \sum_{k=1}^{b} \left[ log(1 - D(G(\bar{x}_j^k))) \right] - \lambda \sum_{k=1}^{bm} \left[ {y_{all}^k}^T log(p(z_{all}^k)) \right] \right\}$

    Update the classifier $\Phi$ and generator $G$ together:

    $\theta_c := \theta_c - \beta \cdot \boldsymbol{grad_c}$ and $\theta_g := \theta_g - \alpha \cdot \boldsymbol{grad_g}$

**end**

---

Table 4: Model attribute classification accuracy (%) on P of PACS-modelset. **Red** and blue indicate the best and second best performance, respectively.

|  | Method | Attributes | | | | | | | | | Avg |
|---|---|---|---|---|---|---|---|---|---|---|---|
|  |  | #act | #drop | #pool | #ks | #conv | #fc | #opt | #bs | #bn |  |
|  | Random | 25.00 | 50.00 | 50.00 | 50.00 | 33.33 | 33.33 | 33.33 | 33.33 | 50.00 | 39.81 |
| P | SVM | 34.20 | 51.70 | 48.50 | 56.10 | 35.70 | 36.50 | 37.60 | 40.50 | 64.60 | 45.04 |
|  | KENNEN* | 37.36 | 53.12 | 57.79 | 59.66 | 38.94 | 35.93 | 37.92 | 41.71 | 63.91 | 47.37 |
|  | SelfReg | 26.08 | 52.35 | 53.89 | 52.70 | 35.11 | 33.84 | 37.46 | 36.42 | 50.99 | 42.09 |
|  | MixStyle | 35.98 | 54.31 | 57.35 | 57.43 | 37.14 | 35.51 | 39.31 | 42.07 | 57.84 | 46.33 |
|  | MMD | 38.67 | 57.16 | 61.49 | 58.73 | 40.65 | 39.14 | 38.69 | 41.06 | 71.48 | 49.67 |
|  | **DREAM** | **39.68** | **57.61** | **64.48** | **60.79** | **40.78** | **40.10** | **43.54** | **43.80** | **72.42** | **51.47** |

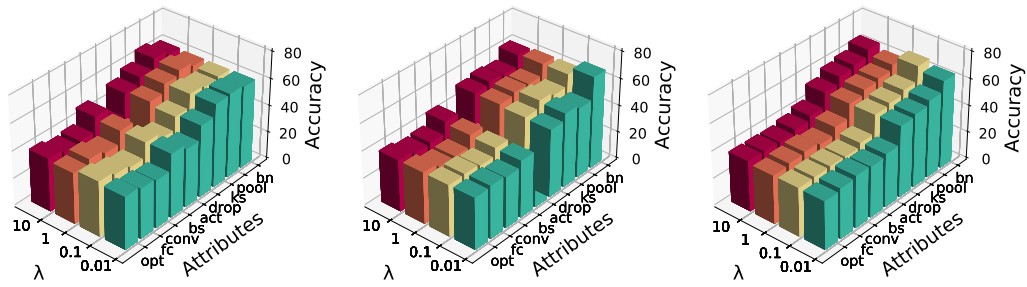

Figure 7: Sensitivity analysis of parameter $\lambda$ on PACS-modelset. From left to right, the results in the P split, C split and S split are shown, respectively.

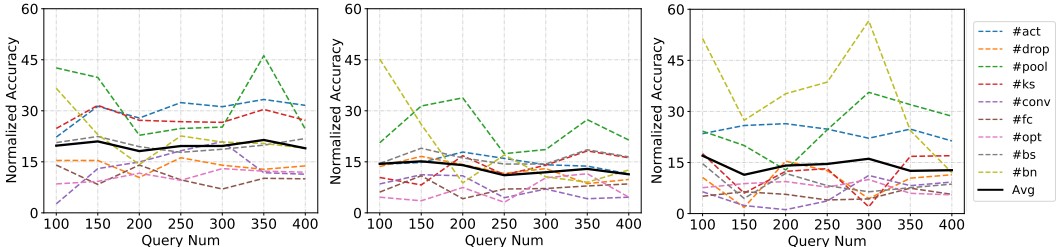

Figure 8: Performance against query number on PACS-modelset. From left to right, normalized accuracies in the P split, C split and S split are shown, respectively.

### A.5 SENSITIVITY ANALYSIS

We study the sensitivity of the trade-off parameter $\lambda$ in our final loss function on the PACS-modelset. As shown in Figure 7, the results for each model attribute do not show evident fluctuation when changing $\lambda$, suggesting that our proposed method is not sensitive to the choices of $\lambda$ in a wide range.

### A.6 QUERY NUMBER AND SIZE OF TRAINING SET ANALYSIS

**Query Number Analysis.** Moreover, we study the performance of DREAM against the number of queries on PACS-modelset. Following (Oh et al., 2018), we use the normalized accuracy that is linearly scaled according to random choice. As shown in Figure 8, with the increase of query numbers, the average performance does not improve but fluctuate, which means more queries do not necessarily provide more information for our DREAM framework.

**Size of Training Set Analysis.** We further study the performance of our method against the size of training set on PACS-modelset. As shown in Figure 9, we observe that the performance slightly fluctuates from size of 1K to 5K, and does not consistently increase when the size increases. We suspect it can be attributed to the difficulty of our problem for domain-free attribute inference of black-box model, and the nature of OOD problem, $i.e.$, the noise level increases as the size of training set grows. It is worth studying further.

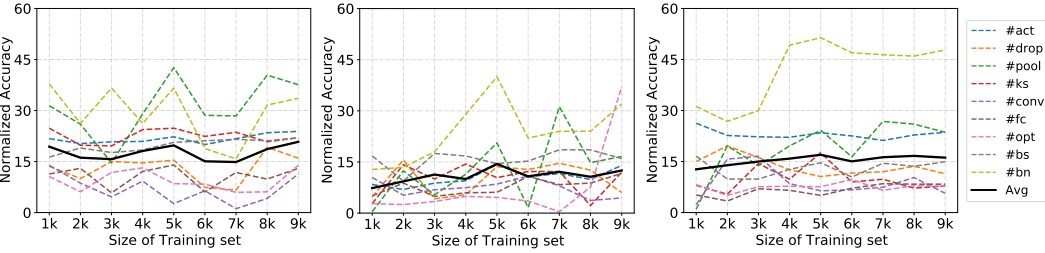

Figure 9: Performance against size of training set on PACS-modelset. From left to right, normalized accuracies in the P split, C split and S split are shown, respectively.

## A.7 STATISTICS OF MODELSET

We represent the statistics of each attribute value in PACS-modelset (Table 5 to Table 7) and MEDU-modelset (Table 8 to Table 11). The "Ratio" line represents the proportion of models with the attribute value in the whole set of models. The next four lines represent maximal, median, mean and minimal accuracy of models for the attribute value, respectively.

Table 5: Distribution of attributes in domain P (Photo) of PACS-modelset and classification accuracy of our method on the Photo validation set.

| | act | | | | drop | | pool | | ks | | conv | | | fc | | |
|---|---|---|---|---|---|---|---|---|---|---|---|---|---|---|---|---|
| | ReLU | ELU | PReLU | Tanh | No | Yes | No | Yes | 3 | 5 | 2 | 3 | 4 | 2 | 3 | 4 |
| Ratio | 24.7 | 25.8 | 24.5 | 25.0 | 50.9 | 49.1 | 50.3 | 49.7 | 50.6 | 49.4 | 33.5 | 33.2 | 33.3 | 33.2 | 32.8 | 34.0 |
| max | 71.3 | 70.9 | 72.8 | 68.7 | 72.8 | 71.3 | 72.4 | 72.8 | 72.8 | 72.4 | 70.5 | 72.8 | 72.4 | 72.8 | 71.1 | 72.4 |
| median | 60.0 | 61.0 | 60.2 | 58.5 | 60.2 | 59.1 | 59.1 | 61.2 | 60.8 | 59.3 | 59.3 | 61.0 | 59.8 | 61.0 | 60.2 | 58.5 |
| mean | 58.0 | 59.7 | 58.4 | 55.7 | 58.2 | 57.7 | 57.9 | 58.0 | 58.6 | 57.3 | 58.2 | 58.6 | 57.0 | 59.6 | 58.3 | 56.1 |
| min | 26.0 | 26.2 | 26.2 | 25.6 | 25.6 | 26.0 | 25.6 | 25.8 | 26.2 | 25.6 | 26.0 | 25.8 | 25.6 | 26.0 | 25.8 | 25.6 |

| | alg | | | bs | | | bn | |
|---|---|---|---|---|---|---|---|---|
| | SGD | ADAM | RMSprop | 32 | 64 | 128 | Yes | No |
| Ratio | 32.4 | 34.0 | 33.7 | 33.6 | 33.9 | 32.6 | 49.8 | 50.2 |
| max | 72.4 | 70.9 | 72.8 | 71.3 | 72.8 | 72.4 | 68.3 | 72.8 |
| median | 56.5 | 60.6 | 61.0 | 61.2 | 60.0 | 57.9 | 56.3 | 63.6 |
| mean | 54.6 | 60.2 | 58.9 | 59.7 | 58.1 | 56.0 | 53.5 | 62.4 |
| min | 25.8 | 26.2 | 25.6 | 26.2 | 25.6 | 25.8 | 25.8 | 25.6 |

Table 6: Distribution of attributes in domain C (Cartoon) of PACS-modelset and classification accuracy of our method on the Cartoon validation set.

| | act | | | | drop | | pool | | ks | | conv | | | fc | | |
|---|---|---|---|---|---|---|---|---|---|---|---|---|---|---|---|---|
| | ReLU | ELU | PReLU | Tanh | No | Yes | No | Yes | 3 | 5 | 2 | 3 | 4 | 2 | 3 | 4 |
| Ratio | 24.9 | 26.6 | 24.9 | 23.6 | 51.0 | 49.0 | 50.7 | 49.3 | 50.6 | 49.4 | 34.5 | 33.4 | 32.1 | 34.3 | 32.9 | 32.3 |
| max | 73.0 | 71.0 | 71.7 | 69.9 | 71.7 | 73.0 | 71.6 | 73.0 | 73.0 | 71.7 | 73.0 | 71.9 | 71.7 | 73.0 | 71.9 | 70.7 |
| median | 62.5 | 63.3 | 62.3 | 61.6 | 62.8 | 62.0 | 61.6 | 63.5 | 63.0 | 61.7 | 60.9 | 63.2 | 63.2 | 63.0 | 62.8 | 61.4 |
| mean | 60.5 | 61.7 | 61.1 | 59.4 | 61.2 | 60.2 | 60.6 | 60.8 | 61.5 | 60.1 | 60.1 | 61.2 | 61.0 | 61.8 | 60.9 | 59.4 |
| min | 25.7 | 25.1 | 26.2 | 25.8 | 25.1 | 25.8 | 26.2 | 25.1 | 25.8 | 25.1 | 25.1 | 25.7 | 27.0 | 25.7 | 25.8 | 25.1 |

| | alg | | | bs | | | bn | |
|---|---|---|---|---|---|---|---|---|
| | SGD | ADAM | RMSprop | 32 | 64 | 128 | Yes | No |
| Ratio | 30.5 | 36.1 | 33.4 | 34.4 | 34.0 | 31.6 | 46.4 | 53.5 |
| max | 71.6 | 71.9 | 73.0 | 73.0 | 71.7 | 71.7 | 70.9 | 73.0 |
| median | 61.0 | 62.8 | 62.9 | 63.0 | 62.3 | 61.6 | 59.9 | 64.5 |
| mean | 59.0 | 61.3 | 61.6 | 61.6 | 60.9 | 59.6 | 58.0 | 63.1 |
| min | 25.1 | 27.2 | 28.8 | 25.8 | 25.8 | 25.1 | 25.1 | 28.4 |

Table 7: Distribution of attributes in domain S (Sketch) of PACS-modelset and classification accuracy of our method on the Sketch validation set.

| | act | | | | drop | | pool | | ks | | conv | | | fc | | |
|---|---|---|---|---|---|---|---|---|---|---|---|---|---|---|---|---|
| | ReLU | ELU | PReLU | Tanh | No | Yes | No | Yes | 3 | 5 | 2 | 3 | 4 | 2 | 3 | 4 |
| Ratio | 25.7 | 27.3 | 25.8 | 21.2 | 51.5 | 48.5 | 49.3 | 50.7 | 50.7 | 49.3 | 34.7 | 33.5 | 31.8 | 34.9 | 32.9 | 32.3 |
| max | 67.4 | 66.5 | 65.0 | 64.8 | 67.4 | 65.6 | 67.4 | 66.5 | 66.5 | 67.4 | 66.5 | 64.9 | 67.4 | 65.1 | 67.4 | 66.5 |
| median | 55.8 | 56.3 | 55.8 | 54.0 | 56.9 | 54.2 | 54.7 | 56.9 | 56.4 | 54.8 | 53.5 | 56.1 | 57.3 | 56.0 | 55.9 | 55.1 |
| mean | 53.9 | 54.4 | 53.8 | 52.0 | 54.9 | 52.2 | 52.5 | 54.7 | 54.6 | 52.6 | 52.0 | 53.9 | 55.1 | 56.0 | 53.7 | 52.8 |
| min | 25.1 | 25.2 | 25.1 | 25.1 | 25.1 | 25.1 | 25.1 | 25.1 | 25.1 | 25.1 | 25.1 | 25.1 | 25.1 | 25.1 | 25.5 | 25.1 |

| | alg | | | bs | | | bn | |
|---|---|---|---|---|---|---|---|---|
| | SGD | ADAM | RMSprop | 32 | 64 | 128 | Yes | No |
| Ratio | 32.0 | 35.8 | 32.2 | 34.3 | 33.6 | 32.1 | 46.0 | 54.0 |
| max | 63.5 | 67.4 | 66.5 | 67.4 | 65.7 | 65.8 | 64.8 | 67.4 |
| median | 54.0 | 56.9 | 55.6 | 57.1 | 54.9 | 54.7 | 55.1 | 56.3 |
| mean | 51.5 | 55.6 | 53.4 | 55.2 | 52.8 | 52.7 | 53.6 | 51.5 |
| min | 25.1 | 25.1 | 25.1 | 25.1 | 25.1 | 25.1 | 25.1 | 25.1 |

Table 8: Distribution of attributes in domain M (MNIST) of MEDU-modelset and classification accuracy of our method on the MNIST validation set.

| | act | | | | drop | | pool | | ks | | conv | | | fc | | |
|---|---|---|---|---|---|---|---|---|---|---|---|---|---|---|---|---|
| | ReLU | ELU | PReLU | Tanh | No | Yes | No | Yes | 3 | 5 | 2 | 3 | 4 | 2 | 3 | 4 |
| Ratio | 24.7 | 25.8 | 24.5 | 25.0 | 50.9 | 49.1 | 50.3 | 49.7 | 50.6 | 49.4 | 33.5 | 33.3 | 33.2 | 33.2 | 32.8 | 34.0 |
| max | 99.3 | 99.2 | 99.3 | 99.2 | 99.3 | 99.3 | 99.3 | 99.2 | 99.2 | 99.3 | 99.1 | 99.2 | 99.3 | 99.3 | 99.2 | 99.3 |
| median | 98.6 | 98.6 | 98.6 | 98.4 | 98.5 | 98.6 | 98.5 | 98.6 | 98.5 | 98.6 | 98.4 | 98.6 | 98.6 | 98.3 | 98.2 | 98.3 |
| mean | 98.4 | 98.5 | 98.4 | 98.3 | 98.3 | 98.5 | 98.3 | 98.4 | 98.3 | 98.5 | 98.4 | 98.6 | 98.6 | 98.6 | 98.5 | 98.5 |
| min | 91.6 | 94.3 | 36.9 | 73.5 | 36.9 | 63.8 | 63.8 | 36.9 | 63.8 | 36.9 | 63.8 | 91.6 | 36.9 | 92.8 | 63.8 | 36.9 |

| | alg | | | bs | | | bn | |
|---|---|---|---|---|---|---|---|---|
| | SGD | ADAM | RMSprop | 32 | 64 | 128 | Yes | No |
| Ratio | 32.4 | 34.0 | 33.6 | 33.5 | 33.9 | 32.5 | 49.7 | 50.3 |
| max | 99.2 | 99.3 | 99.2 | 99.3 | 99.3 | 99.2 | 99.1 | 99.3 |
| median | 98.4 | 98.6 | 98.5 | 98.6 | 98.5 | 98.5 | 98.4 | 98.7 |
| mean | 98.1 | 98.5 | 98.4 | 98.5 | 98.5 | 98.5 | 98.2 | 98.6 |
| min | 36.9 | 92.8 | 90.0 | 90.0 | 92.8 | 36.9 | 36.9 | 63.8 |

Table 9: Distribution of attributes in domain E (EMNIST) of MEDU-modelset and classification accuracy of our method on the EMNIST validation set.

| | act | | | | drop | | pool | | ks | | conv | | | fc | | |
|---|---|---|---|---|---|---|---|---|---|---|---|---|---|---|---|---|
| | ReLU | ELU | PReLU | Tanh | No | Yes | No | Yes | 3 | 5 | 2 | 3 | 4 | 2 | 3 | 4 |
| Ratio | 24.7 | 25.9 | 24.5 | 25.0 | 51.0 | 49.0 | 50.3 | 49.7 | 50.6 | 49.4 | 33.6 | 33.2 | 33.2 | 33.2 | 32.8 | 34.0 |
| max | 99.6 | 99.6 | 99.5 | 99.4 | 99.5 | 99.6 | 99.6 | 99.6 | 99.6 | 99.6 | 99.5 | 99.6 | 99.6 | 99.5 | 99.6 | 99.6 |
| median | 99.1 | 99.1 | 99.1 | 98.9 | 99.1 | 99.0 | 99.0 | 99.1 | 99.0 | 99.1 | 98.9 | 99.1 | 99.1 | 99.1 | 99.1 | 99.0 |
| mean | 98.7 | 98.9 | 98.8 | 98.5 | 98.8 | 98.7 | 98.8 | 98.7 | 98.6 | 98.8 | 98.6 | 98.8 | 98.7 | 98.9 | 98.8 | 98.5 |
| min | 35.7 | 87.8 | 31.9 | 31.9 | 31.9 | 33.4 | 79.2 | 31.9 | 31.9 | 31.9 | 94.5 | 40.6 | 31.9 | 94.7 | 41.3 | 31.9 |

| | alg | | | bs | | | bn | |
|---|---|---|---|---|---|---|---|---|
| | SGD | ADAM | RMSprop | 32 | 64 | 128 | Yes | No |
| Ratio | 32.3 | 34.0 | 33.7 | 33.6 | 34.0 | 32.5 | 49.7 | 50.3 |
| max | 99.6 | 99.6 | 99.5 | 99.5 | 99.6 | 99.5 | 99.5 | 99.6 |
| median | 98.7 | 99.1 | 99.1 | 99.1 | 99.1 | 99.0 | 98.9 | 99.1 |
| mean | 98.2 | 99.0 | 99.0 | 99.0 | 98.8 | 98.4 | 98.4 | 99.0 |
| min | 31.9 | 96.9 | 79.2 | 79.2 | 80.0 | 31.9 | 31.9 | 93.8 |

Table 10: Distribution of attributes in domain D (DIDA) of MEDU-modelset and classification accuracy of our method on the DIDA validation set.

| | act | | | | drop | | pool | | ks | | conv | | | fc | | |
|---|---|---|---|---|---|---|---|---|---|---|---|---|---|---|---|---|
| | ReLU | ELU | PReLU | Tanh | No | Yes | No | Yes | 3 | 5 | 2 | 3 | 4 | 2 | 3 | 4 |
| Ratio | 24.7 | 26.3 | 24.8 | 24.1 | 51.0 | 49.0 | 50.2 | 49.9 | 50.9 | 49.1 | 34.2 | 33.3 | 32.5 | 33.9 | 32.8 | 33.3 |
| max | 97.8 | 98.1 | 97.9 | 98.0 | 97.8 | 98.1 | 97.1 | 97.6 | 98.6 | 98.1 | 94.0 | 94.4 | 99.1 | 97.9 | 98.0 | 98.1 |
| median | 94.0 | 94.4 | 94.5 | 93.7 | 93.9 | 94.3 | 94.0 | 94.3 | 93.9 | 94.4 | 93.0 | 94.5 | 94.8 | 94.3 | 94.1 | 94.0 |
| mean | 92.7 | 93.1 | 93.2 | 92.0 | 92.8 | 92.7 | 92.9 | 92.6 | 92.3 | 93.2 | 91.6 | 93.2 | 93.6 | 93.2 | 92.7 | 92.3 |
| min | 25.4 | 25.0 | 25.2 | 25.1 | 25.0 | 25.1 | 32.7 | 25.0 | 25.0 | 25.0 | 26.0 | 25.1 | 25.0 | 26.5 | 25.0 | 25.1 |

| | alg | | | bs | | | bn | |
|---|---|---|---|---|---|---|---|---|
| | SGD | ADAM | RMSprop | 32 | 64 | 128 | Yes | No |
| Ratio | 31.7 | 35.4 | 32.9 | 33.9 | 33.9 | 32.1 | 47.6 | 52.4 |
| max | 98.0 | 97.7 | 98.1 | 97.9 | 98.1 | 97.9 | 96.6 | 98.1 |
| median | 93.2 | 94.3 | 94.5 | 94.6 | 94.2 | 93.5 | 93.3 | 95.0 |
| mean | 90.4 | 93.9 | 93.8 | 93.8 | 93.0 | 91.3 | 91.1 | 94.2 |
| min | 25.0 | 68.8 | 43.2 | 26.1 | 25.1 | 25.0 | 25.0 | 43.2 |

Table 11: Distribution of attributes in domain U (USPS) of MEDU-modelset and classification accuracy of our method on the USPS validation set.

| | act | | | | drop | | pool | | ks | | conv | | | fc | | |
|---|---|---|---|---|---|---|---|---|---|---|---|---|---|---|---|---|
| | ReLU | ELU | PReLU | Tanh | No | Yes | No | Yes | 3 | 5 | 2 | 3 | 4 | 2 | 3 | 4 |
| Ratio | 25.1 | 26.2 | 25.0 | 23.7 | 51.0 | 49.0 | 49.1 | 50.9 | 50.9 | 49.1 | 33.5 | 33.4 | 33.1 | 33.3 | 32.7 | 34.0 |
| max | 98.9 | 98.9 | 98.8 | 98.7 | 98.8 | 98.9 | 98.9 | 98.8 | 98.8 | 98.9 | 98.5 | 98.7 | 98.9 | 98.9 | 98.8 | 98.9 |
| median | 97.2 | 97.1 | 97.2 | 96.9 | 97.1 | 97.1 | 96.8 | 97.3 | 96.9 | 97.3 | 96.5 | 97.3 | 97.5 | 97.1 | 97.1 | 97.1 |
| mean | 96.5 | 96.3 | 96.6 | 94.9 | 96.4 | 95.8 | 95.3 | 96.8 | 95.8 | 96.4 | 94.8 | 96.5 | 97.0 | 96.2 | 96.3 | 95.8 |
| min | 30.4 | 30.5 | 29.6 | 26.1 | 26.1 | 26.7 | 26.7 | 26.1 | 26.7 | 26.1 | 28.5 | 26.7 | 26.1 | 36.1 | 26.7 | 26.1 |

| | alg | | | bs | | | bn | |
|---|---|---|---|---|---|---|---|---|
| | SGD | ADAM | RMSprop | 32 | 64 | 128 | Yes | No |
| Ratio | 32.9 | 34.8 | 32.3 | 33.5 | 34.0 | 32.5 | 48.7 | 51.3 |
| max | 98.9 | 98.9 | 98.8 | 98.9 | 98.8 | 98.8 | 98.6 | 98.9 |
| median | 96.9 | 97.3 | 97.2 | 97.3 | 97.1 | 96.8 | 98.6 | 98.9 |
| mean | 96.1 | 96.5 | 95.7 | 96.8 | 96.3 | 95.2 | 96.7 | 95.5 |
| min | 26.1 | 29.7 | 26.7 | 31.8 | 26.7 | 26.1 | 26.1 | 26.7 |

## A.8 EXPERIMENT OF MORE REALISTIC SCENARIO

We use a subset of these K classes for training. when training the white box model, we leave out the "dog" and "elephant" for each domain. Then, we train domain-free meta classifiers using model outputs without the "dog" and "elephant" classes and then test domain-free meta classifiers using model outputs that hold all classes. As shown in the 12, our method can still perform well compared with baselines.

Table 12: Model attribute classification accuracy (%) on S of PACS-modelset. **Red** and blue indicate the best and second best performance, respectively. **DREAM\*** represents that its domain-free meta classifier is trained on model outputs without the "dog" and "elephant" classes and is tested on model outputs that hold all classes.

| | Method | Attributes | | | | | | | | | Avg |
|---|---|---|---|---|---|---|---|---|---|---|---|
| | | #act | #drop | #pool | #ks | #conv | #fc | #opt | #bs | #bn | |
| | Random | 25.00 | 50.00 | 50.00 | 50.00 | 33.33 | 33.33 | 33.33 | 33.33 | 50.00 | 39.81 |
| S | SVM | 23.80 | 47.60 | 47.40 | 45.80 | 33.80 | 34.50 | 31.80 | 34.30 | 53.10 | 39.12 |
| | KENNEN* | 34.64 | 50.10 | 53.07 | 52.01 | 34.61 | 37.11 | 35.78 | 37.04 | 55.27 | 43.29 |
| | SelfReg | 27.07 | 54.32 | 51.39 | 53.07 | 36.99 | 36.82 | 35.47 | 34.17 | 61.80 | 43.46 |
| | MixStyle | 37.78 | 51.71 | 54.16 | 53.60 | 34.53 | 36.16 | 36.36 | 36.02 | 59.42 | 44.42 |
| | MMD | 31.96 | 52.94 | 56.84 | 52.78 | 38.18 | 38.20 | 36.20 | 35.92 | 57.56 | 44.51 |
| | **DREAM** | **42.24** | 55.68 | 61.82 | 58.34 | 39.55 | 38.39 | 38.51 | 41.39 | **74.39** | 50.03 |
| | **DREAM\*** | 39.71 | **57.74** | **64.73** | **60.79** | **40.79** | **40.14** | **43.54** | **43.80** | 72.51 | **51.53** |

## A.9 EXPERIMENT OF APPLICATION OF REVERSE ENGINEERING

Let us consider the setting of model extraction. The structure of the target model is unknown. We use an arbitrary random network structure to extract the target model with the method DFME [1], and use the structure inferred by our method to extract the target model. As shown in the above table, the experimental result shows that using the structure inferred by our method obtains better extraction performance, indicating that our findings are significant.

**[1] Kariyappa, S., Prakash, A., & Qureshi, M. K. (2021). Maze: Data-free model stealing attack using zeroth-order gradient estimation. In Proceedings of the IEEE/CVF Conference on Computer Vision and Pattern Recognition (pp. 13814-13823).**

Table 13: Accuracy and normalized accuracy of data-free model extraction methods. The structure of student model has three choices, same to victim, randomly generate ten structures and compute average, predicted by DREAM. Results for "DREAM predict" reflect our support for model extraction task in black box setting.

| Dataset | Victim accuracy(black box model) | Structure of student model | | |
|---------|----------------------------------|----------------|--------------|----------------|
| | | #same to victim | #Random(10) | #DREAM predict |
| MNIST | 86.43% | $68.46\%_{(0.79\times)}$ | $45.88\%_{(0.53\times)}$ | $62.81\%_{(0.73\times)}$ |

