# OpenReview forum: "DREAM: Domain-free Reverse Engineering Attributes of Black-box Model"
_ICLR.cc/2023/Conference — Submitted to ICLR 2023_

### Official Review · Reviewer_W21i · 2022-10-24

**Confidence:** 4
**Correctness:** 4
**Technical Novelty And Significance:** 3
**Empirical Novelty And Significance:** 3
**Recommendation:** 6

**Clarity, Quality, Novelty And Reproducibility:**

There is no problem with the quality of this article. I have no negative comments on the quality of the paper and the clarity of the method description. In general, the proposed DREAM framework is not difficult to reproduce.

**Strength And Weaknesses:**

Strength

The setting that training dataset is not available is practical.

Construction of model dataset by enumerating all possible attribute values and all possible combinations.

The experiments are extensive using a variety of models and data， including comparison with baseline methods.  Analysis on diverse modelsets is insightful.

The proposed OOD generalization learning idea can benefit future research.

Domain-free reverse engineering towards the attributes of black-box model is an exciting and novel research problem.


Weaknesses

The contribution of this paper can be further improved if the potential practical use of the proposed DREAM can be elaborated.
It is not clear to me about the novelty of the MDGAN.


**Summary Of The Paper:**

This paper proposes a method to reverse engineer the attributes of black-box neural networks without knowing the training set of the target model. Specifically, this paper transforms the reverse engineering problem into an out of distribution generalization problem and constructs a DREAM framework to predict the attributes of a black-box model. With MDGAN, the author demonstrates it is able to learn domain invariant features in the scenario of attribute inference of black-box mode.

**Summary Of The Review:**

Although there is room for further improvement in the practical application of the proposed framework, the new setting and novel framework proposed in this paper can provide deep insights into the problem of neural network reverse engineering. The model dataset constructed in this paper and large-scale experiments can also serve as important references for subsequent work. I am generally optimistic about this submission.

---

> ### Author Response · Authors · 2022-11-19
> **Response to Reviewer W21i**
>
> > Q1: The contribution of this paper can be further improved if the potential practical use of the proposed DREAM can be elaborated. It is not clear to me about the novelty of the MDGAN.
>
>
> | Dataset | Victim model accuracy |  |        Structure of student model         |                |
> |:-:|:-:|:-:|:-:|:-:|
> |         |                                  | Same to victim             | Random    | DREAM          |
> | MNIST   | 86.43%                           | 68.46% (0.79x)             | 45.88% (0.53x) | 62.81% (0.73x) |
>
> A1: Thanks for your suggestion. We conduct an experiment to demonstrate a practical use case. Let us consider the setting of a model extraction. The structure of the target model is unknown. We use an arbitrary random network structure to extract the target model with the method [1], and use the reversely restored structure to extract the target model. As shown in the above table, we get 62.81% extraction accuracy when we use the architecture predicted by DREAM, which outperforms random architecture and close to the architecture that same as victim. This indicates that our works can benefit the model extraction task. We have added these in the Appendix A.9 of the revised version.
>
> For the novelty of MDGAN, it learns invariant features from probability representations, which is different from previous works that learn invariant features from image representations. MDGAN leverages one generator and multiple discriminators to force the data of different domains to move towards each other, such that the invariant features among different domains can be learned.
>
> [1] Kariyappa, S., Prakash, A., \& Qureshi, M. K. (2021). Maze: Data-free model stealing attack using zeroth-order gradient estimation. In Proceedings of the IEEE/CVF Conference on Computer Vision and Pattern Recognition (pp. 13814-13823).

---

> ### Author Response · Authors · 2022-11-22
> **Discussion Reminder**
>
> We sincerely thank you for your efforts to review our manuscript. We gently remind the reviewer that we tried our best to address your concerns via our responses and revision of the manuscript. We would be delighted to hear more from you if there are any further concerns.

---

### Official Review · Reviewer_p3er · 2022-10-24

**Confidence:** 4
**Correctness:** 4
**Technical Novelty And Significance:** 3
**Empirical Novelty And Significance:** 2
**Recommendation:** 6

**Clarity, Quality, Novelty And Reproducibility:**

**Clarity**: Average. I had trouble with some parsing the exact input/outputs to different blocks in the model (see Concern 3).

**Quality**: Slightly below average, because I am not entirely convinced with the significance of findings (see concerns 1, 2).

**Novelty**: Good. The technical insight (domain generalization) used by the authors is well-motivated, and perhaps will motivate future attack studies (where lack of data can be circumvented by treating it as a domain generalization problem)

**Reproducibility**: Average. Although the code is not provided, I am reasonably confident in reproducing the results given the description in Appendix A1 and A2. One detail that is not clear to be is the architecture of the "reverse model".

**Strength And Weaknesses:**

### Strengths

**1. Evaluation and Results**
- I appreciate the manner in which the approach is evaluated -- specifically, the paper considers numerous OOD baselines along with Kennen (Oh et al., '18). Many of the proposed baselines themselves exhibit marginal improvements.

**2. Insight to Approach**
- The approach uses an intuitive strategy: to treat generalizing to new test-time inferences as a domain-shift problem in the meta-classifiers input space. The authors also present an experiment (Figure 5) to validate that the domain-shift is addressed.

### Concerns

**1. Proposed threat model makes marginally weaker assumptions**
- The paper does take a step in the right direction, by showing that the reverse-engineering attribute attack is possible in spite of the attacker not possessing the victim's dataset. However, if I understand right, the attacker  needs to nonetheless possess a fully labeled dataset, with multi-domain images per label. Moreover, this dataset needs to be large enough to train thousands of different models. As a result, it appears that the paper somewhat strong assumptions nonetheless.

**2. Consequences of reverse engineering attributes**
- Another concern I have is the significance of findings. Although reverse-engineering attributes highlights that confidential IP can be uncovered, it is much weaker in light of e.g., recent model extraction attacks. This is further exacerbated with mediocre improvements over random guessing -- while it is definitely an improvement over previous works, it appears far from perfect (e.g., ~48-57% accuracy, compared to ~40% random accuracy). Could the authors highlight the significance of the findings?
- One potential solution I see is exploiting this information as a prior for complementary attacks. For instance, crafting adversarial examples as show in Oh et al., '18.

**3. Writing**
- I found a few paragraphs and figures difficult to follow and would appreciate if the authors revised them.
- (a) Notation (esp. in Sec. 3.3): it took me many reads to understand what exactly are the outputs $O^i_j$ e.g., "$f_i$ consists of $K$ models of $i$-th domain - what are the $K$ models? how many are domains are there?
- (b) Sec 3.5: Is there a single reverse classifier $\Phi$? (I thought there is a separate classifier for each attribute?) What is the architecture of $\Phi$? Is the input $z^i$ to the classifier a $R^{K \times d'}$ tensor (with $K$ = 5000?)?
- (c) Fig. 3: In the left yellow block, the multi-domain query (3 images) is input to each of the domain models? Rather, I think each of the domain classifiers has inputs only from a single domain?
- (d) (Nitpicks) Numerous typos: "When feed them", "in the scenery of complicated data", ... Please make another pass over the paper.

**Summary Of The Paper:**

- The paper tackles the problem of reverse-engineering attributes (e.g., #conv. layers? Optimizer?) of a black-box model.
- The main motivation is that existing work (Oh et al., '18) assumes the attacker has the same labeled dataset as that used to train the victim black-box model. The paper proposes an approach that overcomes this limitation by using OOD techniques. Specifically, the proposed approach learns a domain-invariant embedding which is subsequently input to the attribute classifier.
- Results are validated on two datasets (PACS and MEDU) and compared with Oh et al. '18 along with other relevant OOD baseline approaches.

**Summary Of The Review:**

I find that the paper advances the state of the black-box attribute reverse-engineering problem, by using a reasonable technical insight and further demonstrating reasonable performance gains over baselines. However, I am not entirely convinced how significant the findings are, given reasonably strong assumptions (e.g., fully labeled data from multiple domains) and the relevance (how useful is ~50% accuracy in determining simple attributes, when random guessing is ~40% accurate).

---

> ### Author Response · Authors · 2022-11-19
> **Response to Reviewer p3er part2**
>
> > Q3: I found a few paragraphs and figures difficult to follow and would appreciate if the authors revised them.
> **(a)** Notation (esp. in Sec. 3.3): it took me many reads to understand what exactly  are the outputs $O_j^i$
>  e.g., " $f_i$ consists of $K$ models of $i$-th domain - what are the models? how many are domains are there?
> **(b)** Sec 3.5: Is there a single reverse classifier $\Phi$? (I thought there is a separate classifier for each attribute?) What is the architecture of $\Phi$? Is the input $z^i$ to the classifier a $R^{K\times d'}$ tensor (with $K$ = 5000?)
> **(c)** Fig. 3: In the left yellow block, the multi-domain query (3 images) is input to each of the domain models? Rather, I think each of the domain classifiers has inputs only from a single domain?
> **(d)** (Nitpicks) Numerous typos: "When feed them", "in the scenery of complicated data", ... Please make another pass over the paper.
>
> A3: **(a)** Sorry for confusing you. Each $f^i$ contains 5,000 models, these models belong to $i$-th domain, and the number of domains is $m$. Then, we input each query $q_i \in Q$ into all models of $i$-th domain to get an output $O^i_j \in R^{K\times C}$, where $O^i_j $ represents $K$ outputs of $i$-th domain for a query.
> We obtain $O^i \in R^{K\times CN}$ by concatenating $N$ outputs .
>
> **(b)** Your understanding is right. We put several Sigmoid function in the last network layer, where each Sigmoid function is used for an attribute.  As for the architecture of $\Phi$, we have added it in Appendix A.2 of the revised version. Besides, the input $z^i$ to the classifier is indeed a $R^{K\times d'}$ tensor, but we use the mini-batch training strategy. Therefore, in the training phase, $K=batchsize$.
>
> **(c)** Sorry for confusing you. The multi-domain query (3 domains) is indeed the input to each of the domain models. To ensure the consistency of the training and testing phase, we prefer to use the multi-domain query, because if we only input the single-domain query to its corresponding model at the training phase, MDGAN and meta-classifier will learn to fit the outputs from these single-domain queries. However, when we infer the attribute of a model with an unseen domain, we cannot know any information about the domain queries. Even though we input the multi-domain query into the model with an unseen domain, it leads that our model cannot generalize well on the model outputs generated by the query.
>
> **(d)** Thanks for your comments. We have corrected the typos in the revised version.
>
> [1] Kariyappa, S., Prakash, A., \& Qureshi, M. K. (2021). Maze: Data-free model stealing attack using zeroth-order gradient estimation. In Proceedings of the IEEE/CVF Conference on Computer Vision and Pattern Recognition (pp. 13814-13823).

---

> ### Author Response · Authors · 2022-11-19
> **Response to Reviewer p3er part1**
>
> > Q1: The paper does take a step in the right direction, by showing that the reverse-engineering attribute attack is possible in spite of the attacker not possessing the victim's dataset. However, if I understand right, the attacker needs to nonetheless possess a fully labeled dataset, with multi-domain images per label. Moreover, this dataset needs to be large enough to train thousands of different models. As a result, it appears that the paper somewhat strong assumptions nonetheless.
>
> A1: Thanks a lot for your comments. To address your concern, we have added an experiment for a more realistic distribution shift. Specifically, when training the white box model, we leave out the “dog” and “elephant” for each domain. Then, we train domain-free meta classifiers using model outputs without the “dog” and “elephant” classes and then test domain-free meta classifiers using model outputs that hold all classes.
>
> | | method |   |   |   |  | Attributes | |  |  |  | Avg |
> | -------- | ------- | ---- | ----- | ----- | ----- | ----- | ----- | ----- | ----- | ----- | ---- |
> |      |      | #act     | #drop     | #pool     | #ks     | #conv   | #fc     | #opt     | #bs     | #bn     |      |
> |     |  Random   | 25.00     | 50.00     | 50.00   | 50.00    |  33.33    | 33.33     | 33.33      | 33.33      | 50.00    | 39.81    |
> |     |    SVM | 23.80    | 47.60     | 47.40     | 45.80     |33.80     | 34.50    | 31.80     | 34.30    | 53.10     | 39.12     |
> |     |   KENNEN*   | 34.64    | 50.10     | 53.07     | 52.01     | 34.61     | 37.11     | 35.78     | 37.04     | 55.27     | 43.29     |
> |     |  SelfReg   | 27.07     | 54.32     | 51.39     | 53.07     | 36.99     | 36.82     | 35.47     | 34.17     | 61.80     | 43.46     |
> |      |  MixStyle    | 37.78    | 51.71     | 54.16     | 53.60     | 34.53     | 36.16     | 36.36     | 36.02     | 59.42     | 44.42     |
> |      |   MMD   | 31.96    | 52.94    | 56.84     | 52.78   | 38.18     | 38.20    | 36.20     | 35.92    | 57.56     | 44.51     |
> |     |  DREAM   | 42.24     | 55.68     | 61.82   | 58.34     | 39.55     | 38.39    | 38.51     | 41.39    | 74.39     | 50.03     |
> |      | DREAM*     | 39.71    |57.74    | 64.73     | 60.79     | 40.79     | 40.14     | 43.54     | 43.80     | 72.51     | 51.53     |
>
> DREAM* conforms the setting of leaving out certain classes. The white-box models are trained with only five classes (except dog and elephant), and are tested on all classes. As shown in the above table, our method can still outperform the baselines. We have added these results in Appendix A.8 of the revised version.
>
> What is more, we conduct another experiment, as shown in Figure 9 in appendix. In the case that we only use 1,000 models as the training set, the accuracy of attribute inference is still high. So it is expected that our method does not require pretraining many models, and we can obtain satisfactory accuracy with a relatively small number of models.
>
> > Q2: Another concern I have is the significance of findings. Although reverse-engineering attributes highlights that confidential IP can be uncovered, it is much weaker in light of e.g., recent model extraction attacks. This is further exacerbated with mediocre improvements over random guessing -- while it is definitely an improvement over previous works, it appears far from perfect (e.g., ~48-57\% accuracy, compared to ~40\% random accuracy). Could the authors highlight the significance of the findings?
>
> | Dataset | Victim model accuracy |  |        Structure of student model         |                |
> |:-:|:-:|:-:|:-:|:-:|
> |         |                                  | Same to victim             | Random     | DREAM          |
> | MNIST   | 86.43%                           | 68.46% (0.79x)             | 45.88% (0.53x) | 62.81% (0.73x) |
>
>
> A2: Thanks for your insightful comments. To address this concern, we carried out an additional experiment. Let us consider the setting of model extraction. The structure of the target model is unknown. We use an arbitrary random network structure to extract the target model with the method DFME [1], and use the structure inferred by our method to extract the target model.
>
> As shown in the above table, we get 62.81% extraction accuracy when we use the architecture predicted by DREAM, which outperforms random architecture and close to the architecture that same as victim. This indicate that our findings are significant. We have added these results in Appendix A.9 of the revised version. Moreover, there are few works on reverse engineering. We expect that our work can motivate more research efforts to explore this problem, and finally, practical application performance can be achieved.

---

> ### Author Response · Authors · 2022-11-22
> **Discussion Reminder**
>
> We sincerely thank you for your efforts to review our manuscript. We gently remind the reviewer that we tried our best to address your concerns via our responses and revision of the manuscript. We would be delighted to hear more from you if there are any further concerns.

---

### Official Review · Reviewer_iGeV · 2022-10-25

**Confidence:** 4
**Clarity, Quality, Novelty And Reproducibility:** 1. The proposed method has a limited …
**Correctness:** 2
**Technical Novelty And Significance:** 1
**Empirical Novelty And Significance:** 2
**Recommendation:** 3

**Strength And Weaknesses:**

The DREAM problem is somehow interesting. The empirical support is sufficient. However, I have the following concerns:

Weakness:
1) This work has limited technical contributions. The proposed method is simply an extension of adversarial domain adaptation (ADA) [1], which also learns representations invariant to the domains via adversarial training for the downstream task. The authors may argue the difference between a single discriminator and multiple discriminators, which is actually a marginal modification. Even, the multiple discriminators require larger computation. I think it is more efficient to devise a single discriminator with multi-classification on domain labels.

2) The technical part exists severe mistakes. The domain-free reverse classifier should target the classification on attributes of the model instead of the data labels. Eq. (4) and Eq. (5) are incorrectly defined.

3) The OOD generalization only considers the domain shift, which is somehow restricted. As there is no access to the target model’s dataset, the training data may deviate significantly from the target dataset, such as having completely different labels than the desired dataset. Exploring this would be more interesting.

4) There is no introduction to the OOD baselines. I think the authors should also analyze the challenges or issues when directly applying the OOD baselines to the target problem from technical principles in addition to empirical support.

5) Some experimental setups/analyses/results are confusing.
I. What do you mean “We also sample 5000, 1000, 1000 from white-box models as the training set, validation set, and testing set”.
II. “In the scenery of easier cases, e.g., #act, #ks, #fc in M of MEDU (as shown in Table 3), our proposed DREAM is more likely to overfit due to more parameters, degrading the performance of our method.” How to understand it? What is “more parameters” meaning?
III. T-SNE of DREAM looks a bit strange and seems unstructured, i.e., “a long line”.

References
[1] Ganin, Y., Ustinova, E., Ajakan, H., Germain, P., Larochelle, H., Laviolette, F., ... & Lempitsky, V. (2016). Domain-adversarial training of neural networks. The journal of machine learning research, 17(1), 2096-2030.


**Summary Of The Paper:**

This paper targets a new setting of domain-free reverse engineering the attributes of black-box models (DREAM) and casts it as an out of distribution (OOD) generalization problem. In particular, a multi-discriminator generative adversarial network (MDGAN) is proposed to learn domain invariant features on the training data that consists of multiple domains. Then a domain-free reverse model is learnt on domain invariant features, which can be generalized to infer the attributes of black-box model with a new domain. Extensive experimental studies are conducted to compare the proposed method with various baselines.

**Summary Of The Review:**

The DREAM problem is somehow interesting. The empirical support is sufficient. However, I have concerns about the novelty and correctness of the techniques. In addition, I think the conducted case is somehow restricted and the technical motivation of this work is not significantly strong.

---

> ### Author Response · Authors · 2022-11-19
> **Response to Reviewer  iGeV part3**
>
> > Q5: Some experimental setups/analyses/results are confusing. I. What do you mean “We also sample 5000, 1000, 1000 from white-box models as the training set, validation set, and testing set”. II. “In the scenery of easier cases, e.g., \#act,\#ks, \#fc in M of MEDU (as shown in Table 3), our proposed DREAM is more likely to overfit due to more parameters, degrading the performance of our method.” How to understand it? What is “more parameters” meaning? III. T-SNE of DREAM looks a bit strange and seems unstructured, i.e., “a long line”.
>
> A5: I. “We also sample 5000, 1000, 1000 from white-box models as the training set, validation set, and testing set” means that, for each domain, we sampled 5000, 1000, 1000 from 10,000 pretrained white box models as the training set, test set, and validation set.
>
> II. The sentence “In the scenery of easier cases, e.g., \#act, \#ks, \#fc in M of MEDU (as shown in Table 3), our proposed DREAM is more likely to overfit due to more parameters, degrading the performance of our method.” means that our model has more parameters than other baselines, so overfitting may occur on attributes such as \#act, \#ks, \#fc in M of MEDU, resulting in worse performance than  baselines
>
> As for the T-SNE [4] result, we think it is reasonable, not strange. This is because we attempt to learn invariant features from multiple domains. If we can successfully learn invariant features, all samples of different domains should be not discriminative. Thus, Figure 5 shows our method can indeed learn invariant features.
>
> [1] Kim, D., Yoo, Y., Park, S., Kim, J., \& Lee, J. (2021). Selfreg: Self-supervised contrastive regularization for domain generalization. In Proceedings of the IEEE/CVF International Conference on Computer Vision (pp. 9619-9628).
>
> [2] Zhou, K., Yang, Y., Qiao, Y., \& Xiang, T. (2021). Domain generalization with mixstyle. arXiv preprint arXiv:2104.02008.
>
> [3] Li, H., Pan, S. J., Wang, S., \& Kot, A. C. (2018). Domain generalization with adversarial feature learning. In Proceedings of the IEEE conference on computer vision and pattern recognition (pp. 5400-5409).
>
> [4] Van der Maaten, L., \& Hinton, G. (2008). Visualizing data using t-SNE. Journal of machine learning research, 9(11).

---

> > ### Comment · Reviewer_iGeV · 2022-11-23
> > **Thanks for authors’ responses.**
> >
> > However, my concerns are not fully addressed.
> >
> > 1. I admit that the proposed new problem has some contributions but the contributions is somehow marginal. As pointed out by reviewer sNFK, it is a combination of reverse engineering and adversarial domain generalization.  Thus, I do not fully agree that this work is a distinct solution to reverse engineering. In addition, the authors claim that their work is to learn invariant features from probability representations, which is different from previous works learning invariant features from image representations. From my point of view, they just differ in the input. The technique to learn invariant representations is exactly the same.
> >
> > A single discriminator with multi-classification on domain labels is possible to achieve adversarial domain generalization. Specifically, the discriminator is to classify different domains and the generator is to generate representations that confuse the discriminator.
> >
> > Although the authors justify that ADA only involves one source domain and one target domain, adversarial multiple source domain adaptation has also been well explored before [1]. So I feel this work has limited technical contribution.
> >
> > 2. In Page 5 of the original draft, $C$ is denoted as the number of classes in the dataset, and $y$ is not explained further. Thus, I understand $y$ (its calculation involving $C$) in Eq. (4) and Eq. (5) represents class labels.
> >
> > 3. The challenges or issues when directly applying the OOD baselines to the target problem from technical principles are still not explained well. The difference between image representations and probability representations is too vacant.
> >
> >
> > References
> >
> > [1] Zhao, H., Zhang, S., Wu, G., Moura, J. M., Costeira, J. P., & Gordon, G. J. (2018). Adversarial multiple source domain adaptation. Advances in neural information processing systems, 31.

---

> ### Author Response · Authors · 2022-11-19
> **Response to Reviewer iGeV part2**
>
> >Q3: The OOD generalization only considers the domain shift, which is somehow restricted. As there is no access to the target model’s dataset, the training data may deviate significantly from the target dataset, such as having completely different labels than the desired dataset. Exploring this would be more interesting.
>
> A3: Thanks for your comments.If the labels are completely different, the outputs of the trained white box model and the tested black box model are completely different. We argue that this is infeasible.
> For example, it is difficult to train a white box with the handwritten numeral in MNIST and test a black box with CIFAR10. They can hardly learn the invariant features because they are not a kind of thing at all.
> However, in real-world applications, a black-box model deployed on a cloud platform basically provides its functionality and which categories it can output. Therefore, it is possible to collect data with the same label space but different distributions  for training white-box models.
>
> To increase the difficulty of the dataset, we add an experiment on PACS-modelset. Specifically, when training the white box model, we leave out the “dog” and “elephant” for each domain. Then, we train domain-free meta classifiers using model outputs without the “dog” and “elephant” classes and then test domain-free meta classifiers using model outputs that hold all classes. The results are listed in the following table as:
>
> | | method |   |   |   |  | Attributes | |  |  |  | Avg |
> | -------- | -------- | -------- | -------- | -------- | -------- | -------- | -------- | -------- | -------- | -------- | -------- |
> |      |      | #act     | #drop     | #pool     | #ks     | #conv   | #fc     | #opt     | #bs     | #bn     |      |
> |     |  Random   | 25.00     | 50.00     | 50.00   | 50.00    |  33.33    | 33.33     | 33.33      | 33.33      | 50.00    | 39.81    |
> |     |    SVM | 23.80    | 47.60     | 47.40     | 45.80     |33.80     | 34.50    | 31.80     | 34.30    | 53.10     | 39.12     |
> |     |   KENNEN*   | 34.64    | 50.10     | 53.07     | 52.01     | 34.61     | 37.11     | 35.78     | 37.04     | 55.27     | 43.29     |
> |     |  SelfReg   | 27.07     | 54.32     | 51.39     | 53.07     | 36.99     | 36.82     | 35.47     | 34.17     | 61.80     | 43.46     |
> |      |  MixStyle    | 37.78    | 51.71     | 54.16     | 53.60     | 34.53     | 36.16     | 36.36     | 36.02     | 59.42     | 44.42     |
> |      |   MMD   | 31.96    | 52.94    | 56.84     | 52.78   | 38.18     | 38.20    | 36.20     | 35.92    | 57.56     | 44.51     |
> |     |  DREAM   | 42.24     | 55.68     | 61.82   | 58.34     | 39.55     | 38.39    | 38.51     | 41.39    | 74.39     | 50.03     |
> |      | DREAM*     | 39.71    |57.74    | 64.73     | 60.79     | 40.79     | 40.14     | 43.54     | 43.80     | 72.51     | 51.53     |
>
>
> DREAM* conforms the setting of leaving out certain classes. The white-box models are trained with only five classes (except dog and elephant), and are tested on all classes. As shown in the above table, our method can still outperform the baselines. We have added these results in Appendix A.8 of the revised version.
>
> > Q4: There is no introduction to the OOD baselines. I think the authors should also analyze the challenges or issues when directly applying the OOD baselines to the target problem from technical principles in addition to empirical support.
>
> A4: We have added more descriptions about OOD baselines in the experiment part of the revised version as:
>
> Baseline SelfReg [1]: The idea of learning invariant features is to draw samples of similar categories between all domains closer and samples of different categories further.
>
> Baseline MixStyle [2]: The motivation of MixStyle is that the styles of domain images are very similar. For example, the style information is captured by the CNN bottom layer, and the MixStyle performs style mixing at the layer.
>
> Baseline MMD [3]: MMD adopts maximum mean discrepancy loss between each two domains.
>
> These methods are originally designed for image feature representations, which has not been studied for probability representations so far. In this paper, we attempt to explore how to learn invariant features from multiple domains only based on probability representations.

---

> ### Author Response · Authors · 2022-11-19
> **Response to Review iGeV part1**
>
> > Q1: This work has limited technical contributions. The proposed method is simply an extension of adversarial domain adaptation (ADA) [5], which also learns representations invariant to the domains via adversarial training for the downstream task. The authors may argue the difference between a single discriminator and multiple discriminators, which is actually a marginal modification. Even, the multiple discriminators require larger computation. I think it is more efficient to devise a single discriminator with multi-classification on domain labels.
>
> A1: Thanks for your  comments. We think we  have made sufficient contributions, which are summarized as four folds:
> Firstly, the starting point for us to solve the problem is novel. We constitute the first attempt to study reverse engineering problem on an arbitrary black-box model without requiring to know the training data of the target model.
> Secondly, we cast the task as an out of distribution (OOD) generalization problem, which is a distinct solution to reverse engineering.
> Thirdly, different from previous works learning invariant features from image representations, we successfully learn invariant features from probability representations, which has not been explored before.
> Finally, we provide a new solution to a novel problem, and perform extensive experimental studies demonstrating the effectiveness of our method.
>
> Regarding ADA [5], it aims to solve the Domain Adaptation (DA) task, involving only one source domain and one target domain. But it cannot be applied well to Domain Generalization (DG). The difference between DA and DG is whether information of the target domain can be known in advance (known for DA but unknown for DG). Therefore, when solving the DG problem, multiple (more than two) source domains are used to train a machine learning model. In ADA, The single discriminator is used to confuse features between a source domain and a target domain. However, a single discriminator cannot achieve that. Since multiple real and false domain pairs are input into one discriminator in the training process, this discriminator does not know which domain is real and which is fake. For example, the discriminator takes $<(domain1,real), (domain2,fake), (domain3,fake)>$ as input, and it also takes $<(domain1,fake),(domain2,real), (domain3,fake)>$ as input. Therefore, it is necessary to design the structure of MDGAN.
>
> > Q2: The technical part exists severe mistakes. The domain-free reverse classifier should target the classification on attributes of the model instead of the data labels. Eq. (4) and Eq. (5) are incorrectly defined.
>
> A2: Thanks for your comments. With respect, we cannot agree with you that Eq. (4) and Eq. (5) are incorrectly defined. Indeed, the domain-free classifier should target at the classification on attributes of the model. For Eq. (4), we take a softmax operation to the output from the generator. This generator takes model outputs as input, so its output probability $p(z^i)$ is model related, which corresponds to the model attribute value. For Eq. (5), since the $p(z^i)$ is model related, it is still model related after sending $p(z^i)$ to the domain-free reverse classifier. Therefore,  $y$ is the ground truth of the model attributes value.  To clarify this, we have made modification in Page 6 of the revised version.

---

> ### Author Response · Authors · 2022-11-22
> **Discussion Reminder**
>
> We sincerely thank you for your efforts to review our manuscript. We gently remind the reviewer that we tried our best to address your concerns via our responses and revision of the manuscript. We would be delighted to hear more from you if there are any further concerns.

---

### Official Review · Reviewer_sNFK · 2022-10-27

**Confidence:** 3
**Correctness:** 4
**Technical Novelty And Significance:** 2
**Empirical Novelty And Significance:** 3
**Recommendation:** 6

**Clarity, Quality, Novelty And Reproducibility:**

The novelty is limited (see above). The quality of the writing is good, and the quality/reproducibility of the experiments is good.

**Strength And Weaknesses:**

Strengths:
- A useful extension of (Oh et al., 2018) to a more realistic setting
- The writing is mostly clear and the paper is structured well
- Results show that the proposed DREAM method outperforms the previous method that does not consider the possibility of domain shift in the training set of the black-box model

Weaknesses:
- Limited novelty; this is an X+Y paper where X = the reverse engineering approach from (Oh et al., 2018) and Y = adversarial domain generalization
- The paper claims to be concerned about realism of the setting, but the experiments on PACS run somewhat counter to this. Surely one would know whether a black-box model being stolen is trained on cartoons vs photos, and thus be able to collect a dataset for training the white-box models that consists of the proper domain. A more realistic distribution shift would be leaving out certain modes of the data, or certain classes.
- Serious missing citation: "Domain-Adversarial Training of Neural Networks". This is a well-known domain generalization paper using a similar idea to the one proposed in this paper, so it should be cited and compared to.
- Are the SelfReg, MixStyle, and MMD baselines applied to KENNEN? This isn't clear.


**Summary Of The Paper:**

This paper investigates how to predict attributes of black-box models (e.g., provided through a prediction API). Unlike previous work, the authors investigate the more realistic case where the training set is unknown. Adversarial domain generalization tools are used to learn a domain-free reverse engineering classifier for black-box model attributes. In experiments, this outperforms the previous method that does not consider domain shift.

**Summary Of The Review:**

Due to the limited novelty of the paper, I cannot recommend acceptance at this time. However, I may have misunderstood some aspect of the paper and would be willing to reconsider based on the author response.

-----------
Update after rebuttal:

The authors addressed my technical concerns, but I'm still not entirely convinced of the novelty of the paper. Still, this is a solid improvement/extension over (Oh et al., 2018), so I'm now leaning towards acceptance.

---

> ### Author Response · Authors · 2022-11-19
> **Response to Reviewer sNFK**
>
> > Q1: Limited novelty; this is an X+Y paper where X = the reverse engineering approach from (Oh et al., 2018) and Y = adversarial domain generalization
>
> A1: Thanks for your comments. With respect, we cannot fully agree with you. Firstly, The starting point for us to solve the problem is novel. We constitute the first attempt to study reverse engineering problem on an arbitrary black-box model without requiring to know the training data of the target model. Secondly, we cast the task as an out of distribution (OOD) generalization problem, which is a distinct solution to reverse engineering. Thirdly, different from previous works learning invariant features from image representations, we aim to learn invariant features from probability representations, which has not been explored before. Finally, we provide a new solution to a novel problem and perform extensive experimental studies, which demonstrate the effectiveness of our method.
>
> > Q2: The paper claims to be concerned about realism of the setting, but the experiments on PACS run somewhat counter to this. Surely one would know whether a black-box model being stolen is trained on cartoons vs photos, and thus be able to collect a dataset for training the white-box models that consists of the proper domain. A more realistic distribution shift would be leaving out certain modes of the data, or certain classes.
>
> A2: Thanks for your comments. We have added an experiment for the suggested more realistic distribution shift. Specifically, when training the white box model, we leave out the “dog” and “elephant” for each domain. Then, we train domain-free meta classifiers using model outputs without the “dog” and “elephant” classes and then test domain-free meta classifiers using model outputs that hold all classes.
>
> | | method |   |   |   |  | Attributes | |  |  |  | Avg |
> | -------- | -------- | -------- | -------- | -------- | -------- | -------- | -------- | -------- | -------- | -------- | -------- |
> |      |      | #act     | #drop     | #pool     | #ks     | #conv   | #fc     | #opt     | #bs     | #bn     |      |
> |     |  Random   | 25.00     | 50.00     | 50.00   | 50.00    |  33.33    | 33.33     | 33.33      | 33.33      | 50.00    | 39.81    |
> |     |    SVM | 23.80    | 47.60     | 47.40     | 45.80     |33.80     | 34.50    | 31.80     | 34.30    | 53.10     | 39.12     |
> |     |   KENNEN*   | 34.64    | 50.10     | 53.07     | 52.01     | 34.61     | 37.11     | 35.78     | 37.04     | 55.27     | 43.29     |
> |     |  SelfReg   | 27.07     | 54.32     | 51.39     | 53.07     | 36.99     | 36.82     | 35.47     | 34.17     | 61.80     | 43.46     |
> |      |  MixStyle    | 37.78    | 51.71     | 54.16     | 53.60     | 34.53     | 36.16     | 36.36     | 36.02     | 59.42     | 44.42     |
> |      |   MMD   | 31.96    | 52.94    | 56.84     | 52.78   | 38.18     | 38.20    | 36.20     | 35.92    | 57.56     | 44.51     |
> |     |  DREAM   | 42.24     | 55.68     | 61.82   | 58.34     | 39.55     | 38.39    | 38.51     | 41.39    | 74.39     | 50.03     |
> |      | DREAM*     | 39.71    |57.74    | 64.73     | 60.79     | 40.79     | 40.14     | 43.54     | 43.80     | 72.51     | 51.53     |
>
> DREAM* conforms the setting of leaving out certain classes. The white-box models are trained with only five classes (except dog and elephant), and are tested on all classes. As shown in the above table, our method can still outperform the baselines. We have added these results in Appendix A.8 of the revised version.
> > Q3: Serious missing citation: "Domain-Adversarial Training of Neural Networks". This is a well-known domain generalization paper using a similar idea to the one proposed in this paper, so it should be cited and compared to.
>
> A3: Thanks for your comments. We cite the suggested paper in the revised manuscript and discuss it with our method in the Related Work part: Our work is related to ADA, which is designed for the domain adaptation task. It uses the method of adversarial between a feature extractor and a discriminator to learn domain invariant features. However, we aim to solve a domain generalization problem involving more than two domains. Thus a single discriminator cannot learn domain invariant features between multiple domains, i.e., it is hard to directly apply ADA to our scenario.
>
> > Q4: Are the SelfReg, MixStyle, and MMD baselines applied to KENNEN? This isn't clear.
>
> A4: Sorry for confusing you. They are not applied to KENNEN. We have added more descriptions in the experimental part of the revised version so that the reader can better understand our comparison setting: The three baselines firstly take probabilities representation as input to learn invariant features, and then a MLP is adopted on these features to predict model attributes.

---

> ### Author Response · Authors · 2022-11-22
> **Discussion Reminder**
>
> We sincerely thank you for your efforts to review our manuscript. We gently remind the reviewer that we tried our best to address your concerns via our responses and revision of the manuscript. We would be delighted to hear more from you if there are any further concerns.

---

### Author Response · Authors · 2022-11-19
**General Response**

We thank the reviewers for their insightful comments. We really appreciate that the reviewers thought our work to be: novelty of technical methods **(sNFK, iGev)**, detailed explanation of baseline **(sNFK, iGev)**, practical use of reverse engineering and significance of experimental findings **(W21i, p3er)**.

We have made point-to-point response to the comments of each reviewer and uploaded our revised version. Finally, we once again thank all reviewers for their insightful comments which are very helpful for improving the quality of our paper.

---

### Decision · Program_Chairs · 2023-01-20

**Decision:**

Reject

**Justification For Why Not Higher Score:**

While three reviewers voted for acceptance, they also agreed novelty and technical contributions remained relatively limited. While the paper indeed provides some insight, the limitations suggest that the paper could be stronger.

**Justification For Why Not Lower Score:**

N/A

**Metareview: Summary, Strengths And Weaknesses:**

The authors propose a method for domain-free reverse engineering of neural network attributes (such as the number of convolutional layers). This work extends that of Oh et al., 2019, which focused on identifying attributes of a model when the training set is known. The key idea is to extend the work of Oh with adversarial domain generalization. While the reviewers noted that the setting is practical, they also noted limited novelty relative to previous work.